



# A new algorithm to generate a priori trace gas profiles for the GGG2020 retrieval algorithm

Joshua L. Laughner[1], Sébastien Roche[2*], Matthäus Kiel[1], Geoffrey C. Toon[1], Debra Wunch[2], Bianca C. Baier[3,4], Sébastien Biraud[5], Huilin Chen[6], Rigel Kivi[7], Thomas Laemmel[8**], Kathryn McKain[3,4], Pierre-Yves Quéhé[9], Constantina Rousogenous[9], Britton B. Stephens[10], Kaley Walker[2], and Paul O. Wennberg[11,12]

[1]Jet Propulsion Laboratory, California Institute of Technology, Pasadena, CA, USA
[2]Department of Physics, University of Toronto, Toronto, Canada
[3]Global Monitoring Laboratory, National Oceanic and Atmospheric Administration, Boulder, CO, USA
[4]Cooperative Institute for Research in Environmental Sciences, University of Colorado - Boulder, Boulder, CO, USA
[5]Lawrence Berkeley National Laboratory, Berkeley, CA, USA
[6]Center for Isotope Research, University of Groningen, Groningen, the Netherlands
[7]Space and Earth Observation Centre, Finnish Meteorological Institute, Sodankylä, Finland
[8]Laboratoire des Sciences du Climat et de l'Environnement (LSCE/IPSL), UMR CEA-CNRS-UVSQ, Gif-sur-Yvette, France
[9]Climate and Atmosphere Research Centre (CARE-C), The Cyprus Institute, Nicosia, Cyprus
[10]Earth Observing Laboratory, National Center for Atmospheric Research (NCAR), Boulder, CO, USA
[11]Division of Geological and Planetary Sciences, California Institute of Technology, Pasadena, CA, USA
[12]Division of Engineering and Applied Science, California Institute of Technology, Pasadena, CA, USA
[*]now at: School of Engineering and Applied Sciences, Harvard University, Cambridge, MA, USA
[**]now at: Department of Chemistry, Biochemistry and Pharmaceutical Sciences, University of Bern, Bern, Switzerland

**Correspondence:** Joshua L. Laughner (josh.laughner@jpl.nasa.gov) or Paul O. Wennberg (wennberg@caltech.edu)

**Abstract.** Optimal estimation retrievals of trace gas total columns require prior vertical profiles of the gases retrieved to drive the forward model and ensure the retrieval problem is mathematically well-posed. For well-mixed gases, it is possible to derive accurate prior profiles using an algorithm that accounts for general patterns of atmospheric transport coupled with measured time series of the gases in questions. Here we describe the algorithm used to generate the prior profiles for GGG2020, a new version of the GGG retrieval that is used to analyze spectra from solar-viewing Fourier transform spectrometers, including the Total Carbon Column Observing Network (TCCON). A particular focus of this work is improving the description of $CO_2$, $CH_4$, $N_2O$, HF, and CO in the stratosphere. We show that the revised priors agree well with independent in situ and space-based measurements and improve the total column retrievals.



## 1 Introduction

The Total Carbon Column Observing Network (TCCON) has been in operation since 2004, beginning with its first dedicated instrument in Park Falls, WI, USA (Wunch et al., 2011). Since then, the network has expanded to 29 active sites located around the world. The network provides column average dry mole fractions (DMFs) of numerous gases, including carbon dioxide ($CO_2$), methane ($CH_4$), nitrous oxide ($N_2O$), hydrofluoric acid (HF), and carbon monoxide (CO). These observations have been used to infer or evaluate natural and anthropogenic carbon fluxes (e.g. Yang et al., 2007; Chevallier et al., 2011; Keppel-Aleks et al., 2012; Basu et al., 2013; Fraser et al., 2013; Ott et al., 2015; Peng et al., 2015; Deng et al., 2016; Wang et al., 2016; Feng et al., 2017; Hedelius et al., 2018; Crowell et al., 2019; Babenhauserheide et al., 2020; Dogniaux et al., 2020; Sussmann and Rettinger, 2020; Zhang et al., 2020; Villalobos et al., 2021), to study carbon transport (e.g. Keppel-Aleks et al., 2012; Polavarapu et al., 2016), and to provide ground-truth values for space-based measurements of $CO_2$ and $CH_4$, including the Greenhouse gas Observing Satellites (GOSAT and GOSAT-2, e.g. Butz et al., 2011; Cogan et al., 2012; Schepers et al., 2012; Boesch et al., 2013; Frankenberg et al., 2013; Liu et al., 2013; Oshchepkov et al., 2013; Yoshida et al., 2013; Dils et al., 2014; Inoue et al., 2014; Heymann et al., 2015; Ohyama et al., 2015; Parker et al., 2015; Dupuy et al., 2016; Inoue et al., 2016; Kulawik et al., 2016; Schepers et al., 2016; Liang et al., 2017a; Ohyama et al., 2017; Velazco et al., 2019), TanSat (Yang et al., 2020), the Orbiting Carbon Observatories (OCO-2 and OCO-3, e.g. Liang et al., 2017a, b; Wunch et al., 2017; Kiel et al., 2019), and the Tropospheric Monitoring Instrument (TROPOMI, e.g. Borsdorff et al., 2019; Schneising et al., 2019; Lorente et al., 2021).

The TCCON instruments are solar-viewing Bruker 125HR (high resolution) Fourier transform infrared (FT-IR) spectrometers, which record an interferogram once every few minutes. These interferograms are processed by the GGG software package to provide column average DMFs. Once the interferograms are converted to spectra, the core routine of GGG calculates the expected spectra from a forward model based on a custom linelist and a priori profiles of the absorbing gases with absorption lines in the fitting window. The retrieval calculates a posterior trace gas profile that minimizes the root mean square (RMS) fitting residuals between the forward modeled and observed spectra.

There are two common terms used to describe different approaches towards finding the optimal posterior profile: a "scaling" retrieval or a "profile" retrieval. In a scaling retrieval, the retrieval multiplies the entire prior profile by a single value, finding the scaled version that produces the best agreement with the observed spectrum. In a profile retrieval, each level of the profile can be varied, with the allowed variation constrained by a specific covariance matrix. Compared to a profile retrieval, a scaling retrieval is faster and does not alias spectroscopic or instrument line shape errors into profile shape errors. It is more sensitive to errors in the *shape* of the prior profile compared to a full profile retrieval because it cannot change the shape of the posterior solution (meaning the ratio of DMFs between levels in the profile cannot change). However, it is not affected by a *uniform multiplicative* error in the prior DMFs at all altitudes. That is, if the entire profile under- or over- estimates the true atmospheric DMFs by the same multiplicative factor, a scaling retrieval can—in theory—perfectly correct the retrieved profile. Roche et al. (2021) examines the differences between scaling and profile retrievals in the context of TCCON data in more detail.



The relationship between the shape error in the prior and the error in the retrieved column amount depends on the averaging kernels. For TCCON $CO_2$ retrievals, testing with synthetic spectra shows that a 1% error in the profile shape leads to an error of $\leq 0.025\%$ in $XCO_2$ at solar zenith angles (SZAs) $\lesssim 60°$, and $\leq 0.125\%$ up to SZA $\approx 75°$. (Details of how this was quantified are given in Sect. S1.) This means that for typical SZAs observed by TCCON, an error of 1% to 2% (about 4 to 8 ppm) in the $CO_2$ prior results in a retrieval error well below the 0.25% ceiling required for TCCON data.

In both GGG2014 and GGG2020, the prior profiles are derived as much as possible from meteorological variables and general correlations between these variables and trace gas DMFs in the atmosphere. GGG2014 used meteorological reanalyses from the National Centers for Environmental Prediction (NCEP). GGG2020 uses the Goddard Earth Observing System Forward Product for Instrument Teams (GEOS-5 FP-IT or GEOS FP-IT) reanalysis product. The GEOS FP-IT product was chosen because it is provided on a finer temporal resolution than the NCEP product (3 hourly vs. 6 hourly), is available with a lag of one day in normal operation, and includes diagnosed potential vorticity (PV). The PV fields are of particular importance because they allow the GGG2020 priors to better represent latitudinal transport in the stratosphere, thus improving the stratospheric trace gas profiles. However, GEOS FP-IT data is only available from the year 2000 on, so the GGG package retains the capability to use NCEP meteorology as input data. This capability has been further developed since GGG2014, though we do not include those changes in this paper.

Here, we describe the algorithm used to compute the prior profiles of $CO_2$, $N_2O$, $CH_4$, HF, CO, $H_2O$, and $O_3$ for GGG2020. The algorithm is named "ginput" and is available through GitHub (Laughner, 2022). We begin in this paper by describing the core parts of the algorithm that are common across many of the gases (Sect. 2). We then address elements specific to individual gases in Sect. 3. Finally, we compare the GGG2014 and GGG2020 priors against a wide variety of observations in Sect. 5.

As a final note, the $CO_2$ priors described here are also used in the versions 10 and 11 OCO-2/3 retrievals. There are small differences in the OCO-2/3 priors compared to the TCCON priors which are discussed in Sect. 4.

## 2 General design

The central algorithms for the GGG2020 ($CO_2$, $N_2O$, $CH_4$) priors are similar to each other. Trace gas mole fractions are tied to the monthly average measurements in whole-air flasks sampled at the Mauna Loa, Hawaii (MLO) and American Samoa (SMO) sites operated by the United States National Oceanic and Atmospheric Administration's (NOAA's) Global Monitoring Laboratory. The fundamental underlying assumption of the GGG2020 priors algorithm is that the spatial variation in these gases can be largely captured by accounting for the transport lag between the location of the prior profile and the tropics (where MLO & SMO flask samples are made), and chemistry occurring during stratospheric transport.

The MLO & SMO data used to create the GGG2020 priors ends in December 2018. In order to ensure consistent priors are created with this version of GGG, these files will not be updated until the next GGG release even as NOAA releases more data in the interim. Therefore, it is necessary to extrapolate the MLO & SMO records forward in time for retrievals of spectra taken after December 2018. This is done by:





| Gas | $f(t)$ | $n$ (years) |
|-----|--------|-------------|
| $CO_2$ | $c_0 e^{c_1 t}$ | 10 |
| $CH_4$ | $c_0 + c_1 t + c_2 t^2$ | 5 |
| $N_2O$ | $c_0 + c_1 t + c_2 t^2$ | 10 |

**Table 1.** Function forms ($f(t)$) and number of years used to fit the combined MLO & SMO DMF record to extrapolate beyond 2018. In $f(t)$, the $c$'s are the fit parameters.

1. Fitting a function, $f(t)$ to the last $n$ years of the MLO & SMO records. Both $f(t)$ and $n$ are chosen for each gas to best represent that gas's behavior.

2. Calculating the average seasonal cycle over the last $n$ years as the anomaly relative to $f(t)$.

3. Extend the record to the necessary date using $f(t)$ as the baseline and applying the average seasonal cycle on top of it.

This procedure is shown graphically in Fig. 1.

Details of $f(t)$ and $n$ are provided in Table 1. Note: this method is also used to extrapolate back in time if data prior to the start of the combined MLO & SMO record is needed to represent the distribution of ages of air in the stratosphere (see Sect. 2.3).

Errors in extrapolating the MLO & SMO DMFs will negatively impact the TCCON retrievals if the error in extrapolation introduces an error in the profile shape, due to an El Niño year, for example. Currently, we estimate the error due to extrapolation

to be about 0.25% for $CO_2$, 0.15% for $N_2O$, and 0.6% for $CH_4$ (see S2 in the supplement for details). A scaling retrieval, such as the GGG algorithm used by TCCON, can theoretically perfectly account for an error in magnitude of the prior, thus we deem this level of uncertainty acceptable for TCCON priors. However, a profile retrieval (such as that used by OCO-2/3) will be more negatively impacted by such errors. Therefore, when these priors are used for the version 11 OCO-2/3 retrievals, more recent NOAA data is ingested (see Sect. 4).

Ingesting the MLO & SMO data as the basis for the priors effectively ties those priors to the WMO scale to which the MLO & SMO data are calibrated. Table 2 describes which scale each gas is tied to for each algorithm in which these priors are used. As these priors were developed at the same time as the X2019 $CO_2$ scale (Hall et al., 2021), whether the $CO_2$ priors are tied to the X2007 or X2019 $CO_2$ scale depends on which scale the MLO & SMO data are calibrated to.

Unlike the other gases in Table 2, CO is not tied to its scale through the MLO & SMO data. CO priors are created using

a different approach to the other primary gases; this approach will be described in Sect. 3.6. The relevant point here is that CO is taken from the GEOS FP-IT product (Lucchesi, 2015) and, in the troposphere, is scaled to match observations from the first three Atmospheric Tomography Mission (ATom) aircraft campaigns (Thompson et al., 2022). As the ATom QCLS CO observations used were calibrated to the X2014A scale, the CO priors are considered tied to that scale.



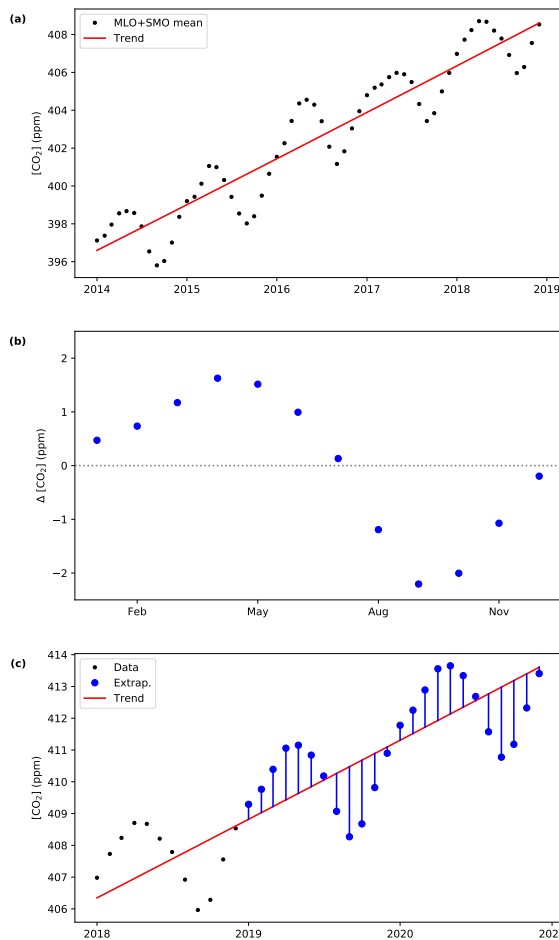

**Figure 1.** Process to extrapolate the combined MLO & SMO monthly average record. **(a)** Fit the last 5 or 10 years with the best function for a given gas. **(b)** Calculate the mean monthly anomaly relative to the trend over the same time period. **(c)** extend the trend in time and apply the mean monthly anomalies on top of it.





| Gas | Scale (GGG2020) | Scale (OCO-2/3 v10) | Scale (OCO-2/3 v11) |
|---|---|---|---|
| $CO_2$ | X2007 | X2007 | X2019 |
| $CH_4$ | X2004 | N/A | N/A |
| $N_2O$ | X2006 | N/A | N/A |
| $CO^*$ | X2014A | N/A | N/A |

**Table 2.** The WMO calibration scales to which the in situ data used in the GGG2020 and OCO-2/3 priors are tied. *Note that, unlike for $CO_2$, $N_2O$, and $CH_4$ (for which this tie comes from the MLO & SMO data), for CO this is from scaling to ATom data in the troposphere.

Several gases (CO, $H_2O$, HDO, $O_3$) are contained in the GEOS-5 FP-IT meteorology product ingested by GGG2020. $H_2O$ and $O_3$ are taken directly from GEOS-5 FP-IT, while CO and HDO are derived from GEOS-5 FP-IT. Details are given in Sect. 3.

Finally there are a large number of gases that must be accounted for as interfering absorbers during retrievals of primary TCCON target gases. These gases use priors derived from climatological profiles from the summer at 35° N. Details are given in Sect. 2.4.

## 2.1 Design rational

In developing the GGG2020 priors, we had two guiding principles in mind:

1. Minimize direct dependence on other measurements or models as much as possible such that retrievals using these priors are independent measurements (in the statistical sense) that other observations or models can be compared to.

2. Produce an algorithm which generates reproducible prior profiles if run at different times.

The first principle is why the GGG2020 priors only ingest MLO & SMO data, rather than more surface data or why we do not use modeled gas profiles (other than for CO). For the much shorter-lived CO, we decided that capturing the spatial variability was worth the trade off of relying on GEOS FP-IT modeled CO (especially as GGG2020 already uses GEOS FP-IT meteorology). Other data used in generating the priors (e.g. latitudinal gradients of $CO_2$ and $CH_4$ from HIPPO & ATom, ACE-FTS profiles) were likewise adopted because the improvement in the priors was deemed worth the loss of statistical independence. Since these data are used to generate static values (such as lookup tables or coefficients in functions) rather than being directly ingested, we retain some independence from these sources.

The second principle is why the GGG2020 priors and OCO-2/3 v10 priors only use MLO & SMO flask data through the end of 2018, rather than updating regularly. One concern raised during development was whether such regular data updates would alter previously obtained data, such as from retrospective quality control. This would introduce a situation where we could not exactly reproduce priors generated using an old version of the input data. Given time constraints, it was not possible to engineer a solution to detect or avoid this issue for GGG2020 and OCO-2/3 v10 priors. With the additional development time for OCO-2/3 v11, we were able to update the priors algorithm to safely ingest more rapidly updated MLO & SMO data.



## 2.2 Tropospheric prior

The GGG2020 tropospheric priors assume that the trend observed by MLO & SMO is driven by emissions in the northern
midlatitudes, thus the measured DMF at MLO & SMO will lag behind the DMFs in the northern hemisphere and precede the
DMFs in the southern hemisphere. To compute the tropospheric DMFs, we average MLO & SMO data together with equal
weight, deseasonalize the MLO & SMO average to get the underlying trend, approximate the offset forward or backward in
time relative to MLO & SMO with an idealized distance function, apply a multiplicative and additive correction to match
observed latitudinal gradients, and impose a latitudinally-dependent seasonal cycle. Mathematically, this follows Eq. (1):

$$130 \quad \mathrm{DMF}(l, z, z_\mathrm{trop}, f_y) = s(l, z, f_y, d) \cdot [\alpha(d) \cdot \mathrm{DMF}_\mathrm{ref}(d) + \beta \cdot l] \tag{1}$$

The variables in this function are:

- $l$ is latitude. In the GGG2020 TCCON priors, this is an "effective latitude" derived from mid-tropospheric potential
  temperature (c.f. Sect. 2.2.1).

- $z$ is altitude with the bottom half of the troposphere stretched downward slightly to treat the bottom layer as being at the
135 surface for the purpose of this calculation (c.f. Sect. 2.2.2).

- $z_\mathrm{trop}$ is the tropopause altitude

- $f_y$ is the fractional year (defined as 1-based day-of-year / 365.25)

- $\mathrm{DMF}_\mathrm{ref}$ is the reference DMF taken from a deseasonalized MLO & SMO trend

- $d$ is the distance offset function, defined by Eq. (2)

- $s$ is the seasonal cycle factor, defined by Eq. (4d)

- $\alpha$ and $\beta$ are coefficients that scale and adjust the ideal gradients assumed by $d$ to account for differences between gases.
  Their values are given in Table 3 and are discussed in detail in Sect. 3.

The distance function $d$ is shown in Fig. 2 (assuming a simple latitudinal dependence for the tropopause altitude). It has the
mathematical form:

$$145 \quad d = d'(l, z; l_\mathrm{ref}, z_\mathrm{trop}) - d'(0°, 0.01\,\mathrm{km}; l_\mathrm{ref}, z_\mathrm{trop}) \tag{2}$$

where

$$d'(l, z; l_\mathrm{ref}, z_\mathrm{trop}) = 0.313 - 0.085 \cdot \exp\left(-\left[\frac{l - l_\mathrm{ref}}{18}\right]^2\right) - 0.268 \cdot \exp\left(-1.42\frac{z}{z + z_\mathrm{trop}}\right) \cdot \frac{l/22}{\sqrt{1 + (l/22)^2}} \tag{3}$$





| Gas | $\alpha$ | $\beta$ |
|---|---|---|
| $CO_2$ | $-3.55 \cdot d(l, z; l_{\text{ref}}, z_{\text{trop}}) \cdot \frac{\partial \text{DMF}_{\text{ref}}}{\partial t}$ | $0$ |
| $N_2O$ | $\exp\left(\frac{-d(l,z;l_{\text{ref}},z_{\text{trop}})}{121 \text{ yr}}\right)$ | $0$ |
| $CH_4$ | $\exp\left(\frac{-d(l,z;l_{\text{ref}},z_{\text{trop}})}{12.4 \text{ yr}}\right)$ | $\begin{cases} 0.75 \text{ ppb/}° & \text{for} \quad l \geq 0 \\ 0 & \text{for} \quad l < 0 \end{cases}$ |

**Table 3.** Values of $\alpha$ and $\beta$ coefficients in Eq. (1) for the three primary well-mixed gases. Rationales for these choices are given in the gas-specific sections (Sect. 3).

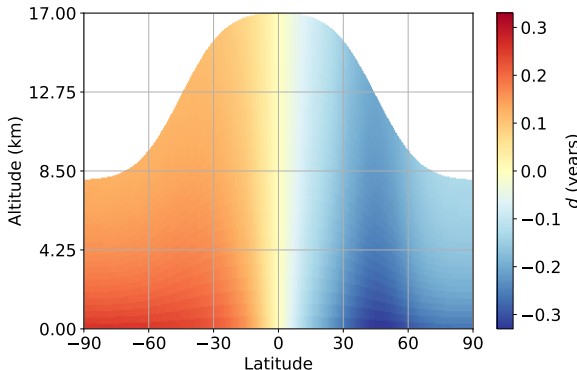

**Figure 2.** Form of the distance function $d$ assuming that emissions occur at $45°$ N and a latitudinally-dependent tropopause height that varies smoothly from 17 km at the equator to 8 km at the poles.

Although $d$ has units of years, it does not represent a physical age or time. It is effectively a basis function to impose the ideal distribution of DMFs relative to MLO & SMO as shown in Fig. 2. Specifically, it assumes that surface DMFs precede

MLO & SMO DMFs in the northern hemisphere, lag MLO & SMO DMFs in the southern hemisphere, and have a smaller latitudinal gradient in the upper troposphere due to faster winds. The basic shape is modified for each gas via $\alpha$ and $\beta$.

$\text{DMF}_{\text{ref}}$ in Eq. (1) is the combined MLO & SMO record, deseasonalized by taking a 12-month rolling mean. This is done because the seasonal cycle at MLO & SMO is not representative of all latitudes. We impose a latitudinally dependent seasonal cycle by multiplying the DMFs by a scaling factor $s$:





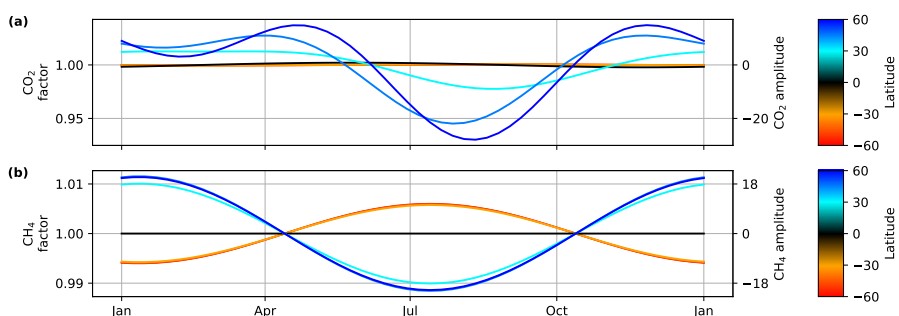

**Figure 3.** Parameterized seasonal cycle for **(a)** $CO_2$ and **(b)** $CH_4$. The left $y$-axis is the factor $s$ in Eq. (4d) and (5d). The right $y$-axis gives what the seasonal cycle amplitude would be for a $CO_2$ DMF of 400 ppm in **(a)** and $CH_4$ DMF of 1800 ppb in **(b)**.

$$s_v = \sin\left(2\pi \cdot [f_y - 0.78]\right) \tag{4a}$$

$$s_l = \frac{s_v \cdot l/15}{\sqrt{1 + (l/15)^2}} \tag{4b}$$

$$s_a = s_l \cdot \exp\left(-d'(l, z; l_{\text{ref}}, z_{\text{trop}})/0.85\right) \tag{4c}$$

$$s = 1 + s_a \cdot c_{\text{gas}} \tag{4d}$$

for all gases but $CO_2$. For $CO_2$ the parameterization is:

$$s_v = \sin\left(2\pi \cdot [f_y - 0.834 - d]\right) \tag{5a}$$

$$s_l = s_v + 1.8 \cdot \exp\left(-\left[\frac{l - 74}{41}\right]^2\right) \cdot (0.5 - s_v^2) \tag{5b}$$

$$s_a = s_l \cdot \exp(-d/0.2) \cdot \left\{1 + 1.33 \cdot \exp\left(-\left[\frac{l - 76}{48}\right]^2\right) \cdot \frac{z + 6}{z + 1.4}\right\} \tag{5c}$$

$$s = 1 + s_a \cdot c_{\text{gas}} \tag{5d}$$

where $f_y$ is the fraction of year passed (defined as 1-based day-of-year / 365.25), $l$ is latitude, $z$ is altitude (in kilometers), $z_{\text{trop}}$ the tropopause altitude (in kilometers), $l_{\text{ref}}$ a reference latitude (45° N), $d'$ is the function from Eq. (3), and $c_{\text{gas}}$ is a gas-specific constant defined in Table S5. $s_v$ represents the basic seasonal variation, $s_l$ the latitudinal variation, $s_a$ the altitude variation. The form of these equations for $CO_2$ and $CH_4$ are shown in Fig. 3.

These parameterized seasonal cycles are the same as that used in GGG2014 priors. The amplitude and phase were derived from surface in situ data and the amplitude is assumed to decay with altitude due to mixing of airmasses with different ages.

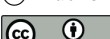



### 2.2.1 Potential temperature-based effective latitude

$CO_2$ profiles for locations on the edge of the tropics are sometimes more "tropical" in nature than their geographic latitudes suggest. In these cases, the observed profile would be more constant versus altitude than the prior profile, which would have some drawdown at the surface.

Keppel-Aleks et al. (2012) showed that, in the extratropics, there is a correlation between 700 hPa potential temperature and $CO_2$ DMFs in the free troposphere, as variations in this potential temperature serve as an indicator of synoptic-scale motion, and therefore the true source latitude of the air. We use dry potential temperature, i.e. the temperature a parcel of dry air would have if brought to a pressure of 1000 hPa adiabatically. This allows us to use potential temperature to derive an "effective latitude" that better predicts the shape of the prior profile. Note that while this was originally developed to improve the $CO_2$ priors, it is used for all gases.

To calculate this effective latitude, we first build a climatology of mid-tropospheric potential temperature from the GEOS-5 FP-IT product by averaging potential temperature between 500 and 700 hPa (henceforth termed $\theta_{mid}$) vs. latitude for two-week periods in 2018 (Fig. 4a). A hypothetical example is shown in Fig. 4b. For a prior in the extratropics, we select the appropriate $\theta_{mid}$-vs.-latitude curve from the table (Fig. 4b, black line) and compare the $\theta_{mid}$ value for the prior against the tabulated mean. If the prior's $\theta_{mid}$ is greater than the mean $\theta_{mid}$ for that latitude, the effective latitude is moved equatorward until it matches, and vice versa if the prior's $\theta_{mid}$ is less.

More specifically, the implementation searches north and south of the prior's geographic latitude for the two latitudes (one north, one south) with the smallest difference between the prior's $\theta_{mid}$ and the mean $\theta_{mid}$. If the difference between the mean $\theta_{mid}$ values at both latitudes is within 0.25 K, then the nearer latitude is used. Otherwise, the latitude with the smallest difference between its $\theta_{mid}$ and the prior's $\theta_{mid}$ is used.

There are two caveats to this approach. First, the effective and true (geographic) latitude must have the same sign—that is, both must be in the same hemisphere. Second, within the tropics (defined as $\pm20°$ of the equator), the effective latitude calculation is disabled and the geographic latitude is used. This is done because mid-tropospheric temperature gradients are weak in the tropics and largely uncorrelated with zonal advection (Sobel et al., 2001). To smoothly blend between geographic and effective latitude, a linear interpolation between them occurs in the $20°$ to $25°$ range. For example, a profile at $22°$ N would have a latitude calculated as $0.6l_g + 0.4l_e$, where $l_g$ is the geographic latitude and $l_e$ the effective latitude.

### 2.2.2 Altitude grid adjustment

The seasonal cycle and distance basis function assume that the surface is at 0 km altitude. To this end, we use an adjusted altitude as $z$ in Eqs. (1) through (5d). To compute this adjusted $z$, we stretch or squeeze the bottom of the altitude grid so that the bottom layer is at the surface altitude from the GEOS-5 FP-IT 2D files. The adjustment follows:

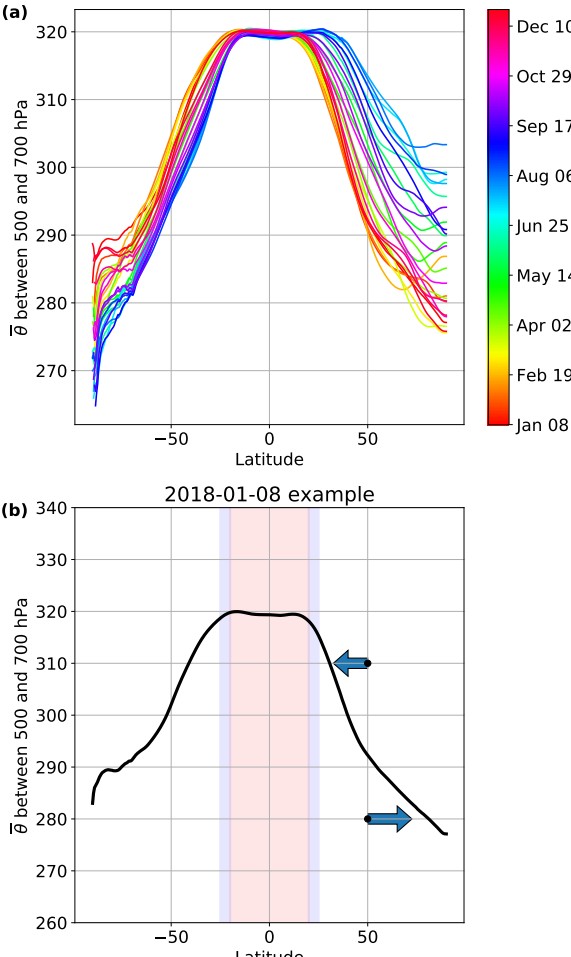

**Figure 4. (a)** the lookup table for $\theta_{\mathrm{mid}}$ vs. latitude and time-of-year. **(b)** a hypothetical example of how the effective latitude calculation works. The black line represents the climatological $\theta_{\mathrm{mid}}$ and the black points represent hypothetical $\theta_{\mathrm{mid}}$ for individual profiles'. The arrows indicate how the effective latitude of each profile is adjusted such that the individual $\theta_{\mathrm{mid}}$ matches the climatological $\theta_{\mathrm{mid}}$. The red shading indicates latitudes where this method is not applied; the blue shading indicates transitional areas between the geographic and effective latitude. (See text for additional details.)





$$
z_{\text{adj}} = \begin{cases} z_{\text{orig}} & \text{if} & z_{\text{orig}} \geq z_{\text{blend}} \\ z_{\text{orig}} + dz \cdot f^2 & \text{if} & z_{\text{min}} \leq z_{\text{orig}} \leq z_{\text{blend}} \\ 0 & \text{if} & z_{\text{orig}} < z_{\text{min}} \end{cases}
\tag{6}
$$

where $z_{\text{orig}}$ is the original altitude, $dz = z_{\text{surf}} - z_{\text{min}}$, $z_{\text{min}}$ is the original grid altitude closest to $z_{\text{surf}}$, $z_{\text{blend}}$ is the original grid altitude closest to $z_{\text{surf}} + \frac{1}{2} \cdot (z_{\text{trop}} - z_{\text{surf}})$, $z_{\text{surf}}$ is the GEOS-5 FP-IT surface altitude, $z_{\text{trop}}$ is the tropopause altitude, and $f$ is:

$$
f = \frac{i_{\text{blend}} - i}{i_{\text{blend}} - i_{\text{min}}}
\tag{7}
$$

where $i_{\text{blend}}$, $i_{\text{min}}$, and $i$ are the indices for $z_{\text{blend}}$, $z_{\text{min}}$, and $z$, respectively. Figure S7 shows an example of the adjustment. This adjustment is minor (typically 50 to 100 m) since the priors are generated on the terrain following levels from the GEOS FP-IT model.

## 2.3 Stratospheric prior

The design of the stratospheric priors draws heavily from Andrews et al. (2001a). That work showed that the profiles of $CO_2$ and $N_2O$ in the lower stratosphere can be well-captured using surface in situ data from the MLO & SMO observatories to determine the trace gas mole fraction entering the stratosphere and then accounting for mixing of air during stratospheric circulation. We extend this method by using atmospheric profile measurements between February 2004 and March 2019 from the Atmospheric Chemistry Experiment Fourier Transform Spectrometer (ACE-FTS, Bernath et al., 2005), data version 3.6 (Boone et al., 2013), to capture chemical production and/or loss of $N_2O$ and $CH_4$ and production of HF.

### 2.3.1 Stratospheric age of air

The age of stratospheric air parcels is calculated from a climatology simulated by the Chemical Lagrangian Model of the Stratosphere (CLaMS) and scaled to match the mean midlatitude age in the Goddard Space Flight Center 2D (GSFC2D) model (Fleming et al., 2011), which provides age of air as a function of latitude, potential temperature, and day-of-year. Age of air in this context refers to the time since the air entered the stratosphere. Figure 5 shows both latitudinal and temporal slices of the CLaMS age of air. The CLaMS model is a 2-D representation of the mean dynamics of the statosphere. To account for the zonal displacements driven by large-scale Rossby waves, we compute an equivalent latitude profile. Equivalent latitude is derived from potential vorticity (PV) following Eq. (1) in Allen and Nakamura (2003).

Note that this equivalent latitude is not the same as the effective latitude used in the tropospheric part of the prior calculation. PV-derived equivalent latitude has been previously shown to predict stratospheric chemical fields well (e.g. Allen and Nakamura, 2003) while a coordinate derived from mid-tropospheric potential temperature predicts synoptic variation in tropospheric trace gas mixing ratios (e.g Keppel-Aleks et al., 2012). Therefore, we use the PV-derived equivalent latitude here for

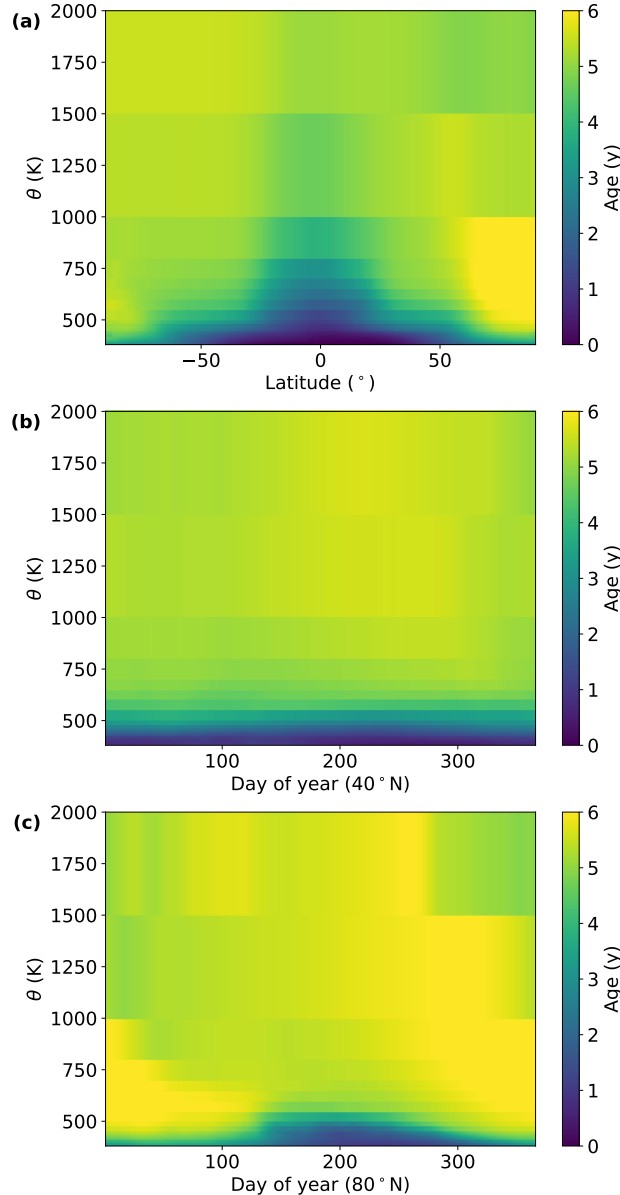

**Figure 5.** Mean age of air from the CLaMS climatology. **(a)** Age vs. latitude and potential temperature for Jan 1, **(b)** Age vs. day of year and potential temperature at $40°$ N, **(c)** As (b), but for $80°$ N.



the stratospheric part of the priors and potential temperature-derived effective latitude in Sect. 2.2.1 for the tropospheric part of the priors.

### 2.3.2 Age spectra and chemistry

Once the age of air is known, we can look backwards in the combined MLO & SMO record to determine the stratosphere boundary condition (SBC), that is, the mole fraction of each gas when a parcel of air entered the stratosphere. The SBC time series is defined as the MLO & SMO average lagged by two months; Andrews et al. (2001a) and references therein show that this is a good proxy for the SBC. However, the mole fraction for a given level in the prior is not simply the mole fraction of, e.g. $CO_2$ when that air entered the stratosphere, but is the result of mixing of air with different ages during convective transport.

This mixing can be represented by solutions to Green's function derived from $CO_2$ measurements (Andrews et al., 2001a), which we represent as age spectra.

Age spectra were precomputed for three regions (tropics, midlatitudes, and polar vortex) and $\sim 45$ different mean ages. Andrews et al. (1999) and Andrews et al. (2001b) showed that different age spectra were necessary to capture tropical and midlatitudinal behavior; likewise, the polar vortex requires its own age spectra form due to strong wintertime descent of air.

Example age spectra are shown in Fig. 6. Note that spectra for the youngest mean ages are not shown.

For each stratospheric level in the priors, the mole fraction of a gas is computed as

$$\overline{c} = \overline{F}(a, \theta) \int S_{a,r}(t) c(t) \, dt \tag{8}$$

where $S_{a,r}(t)$ is the value of the age spectrum for the given mean age ($a$) and region ($r$) and $c(t)$ is the SBC, both at time $t$. That is, the mole fraction is a weighted average of the SBC over time with the weights set by the age spectrum. $\overline{F}(a, \theta)$ is the 245 fraction of gas remaining after chemical loss, $\theta$ is potential temperature, which we use as a vertical coordinate. For $CO_2$ this fraction is always 1, but varies with mean age and potential temperature for other gases, as discussed in more detail in Sect. 3.

### 2.3.3 Middleworld treatment

The middleworld is defined as the part of the atmosphere between the tropopause pressure from GEOS-5 FP-IT and the 380 K isentrope. Of the three tropopause pressure estimates in GEOS-5 FP-IT, we use the blended (thermal and potential vorticity) 250 estimate. The 380 K isentrope is the lowest potential temperature surface entirely contained within the stratosphere; therefore the stratospheric approach described in Sect. 2.3 is only applicable to levels above 380 K (the stratospheric overworld). To fill in the prior in the middleworld, we linearly interpolate mole fraction as a function of potential temperature between the tropopause and 380 K.

### 2.4 Secondary gases

For the purpose of this paper, "secondary gases" are defined as those which are neither tied directly to MLO & SMO records nor the GEOS-5 FP-IT product. This is all gases other than $CO_2$, $N_2O$, $CH_4$, HF, CO, $H_2O$, HDO, and $O_3$. $O_2$ and HCl are



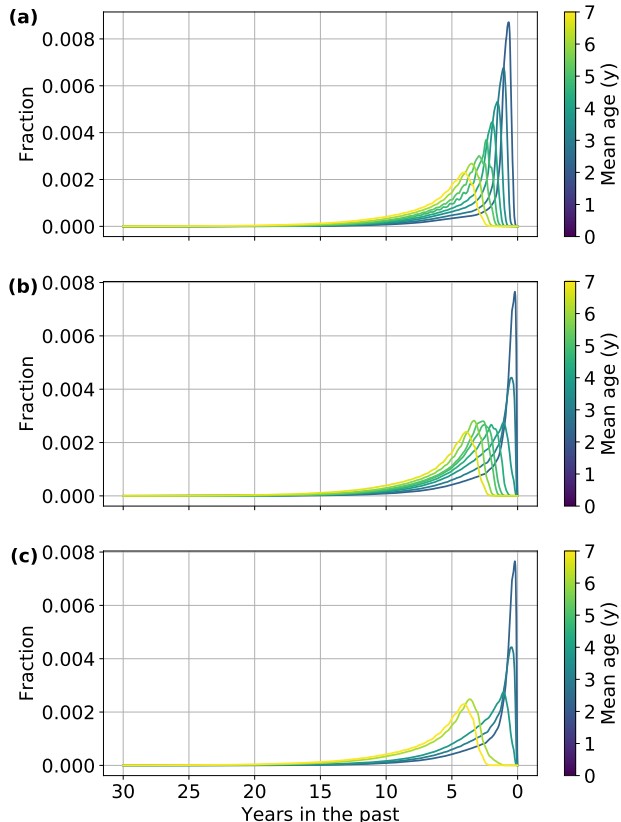

**Figure 6.** Example age spectra for **(a)** tropics, **(b)**, midlatitudes, and **(c)** polar vortex. The $y$-values represent the contribution of air from that time to the average mole fraction of the parcel as a whole. Note that age spectra for the youngest air are not shown because they are nearly delta functions.

the two most relevant to standard TCCON retrievals. Priors for these gases are based on climatological profiles for summer at $35°$ N derived from profiles measured by MkIV spectrometer balloon flights (Toon, 1991) and the ACE-FTS instrument. These climatological profiles are modified for a given location and time in four steps:

1. stretch or compress the profile vertically so that the tropopause is at the correct altitude,

2. apply a latitudinal gradient,

3. apply a secular trend,

4. apply a seasonal cycle.

These steps require the latitude and age of air of the profiles. This approach is nearly identical to that used for all gases in the
GGG2014 priors, except that for steps 2–4, the age of air and effective latitude described in Sect. 2.2 are used in the troposphere





and the CLaMS age and PV-derived equivalent latitude from Sect. 2.3 are used in the stratosphere. The middleworld is filled in by linear interpolation in $\theta$ between the tropopause and 380 K, as is done for the primary gases. Details of the calculation are given in the supplement.

## 2.5 Conversion to number density

All trace gas quantities shown and discussed in this paper are in dry mole fractions (DMFs, i.e. moles of trace gas per moles of dry air). However, in its forward model, GGG uses gas profiles in number density (molec. $cm^{-3}$) for spectroscopic calculations. To convert DMF to number density, we use:

$$n_{gas} = \frac{c_{gas}}{1 + c_{H_2O}} n_{ideal} \tag{9}$$

where $n_{gas}$ is the number density of the gas of interest, $c_{gas}$ is the DMF of that gas, $c_{H_2O}$ is the DMF of water (from the $H_2O$
prior profile), and $n_{ideal}$ the ideal gas number density. The factor $1 + c_{H_2O}$ converts $n_{ideal}$ into number density of dry air.

## 3 Gas-specific design

In this section, we will discuss elements of the algorithm unique to each gas. With the exception of $O_2$, each section will be divided into subsections for the tropospheric and stratospheric priors.

### 3.1 $O_2$

We assume a uniform DMF of 0.2095 for $O_2$ at all altitudes. During the retrieval, this is converted to number density following Sect. 2.5. In the GGG2014 priors, the conversion to number density did not include a correction for water. This led to a profile shape error: as water DMFs are highest near the surface, failing to include the water correction led to an overestimate of the near-surface number density for every absorbing gas.

    The impact of this error in the previous priors on the final column amounts was small because, in public TCCON data, all
gas column amounts are reported as column average mole fractions (termed Xgas, e.g. $XCO_2$). These are calculated as:

$$Xgas = \frac{V_{gas}}{V_{O_2}/0.2095} \tag{10}$$

where $V_{gas}$ and $V_{O_2}$ are the total column amounts (in molec. $cm^{-2}$) of the target gas and $O_2$, respectively. The denominator represents a column of dry air inferred from the retrieved $O_2$ column. The advantage of this method over using a column of air derived from surface pressure is that, because primary TCCON target gases are measured on the same detector as $O_2$, certain
types of instrumental error cancel out in this ratio, reducing their impact on the final data product (Washenfelder et al., 2003; Wunch et al., 2011). Likewise, the shape error due to the missing water correction in GGG2014 priors largely canceled out in the column-averaged Xgas DMFs. However, the GGG2020 treatment, following Eq. (9), is more physically consistent, leads





to more consistent $O_2$ scaling factors retrieved among TCCON stations, and yields a better shape—especially under warm, humid conditions.

## 3.2 $CO_2$

**Troposphere:** The value of $\alpha$ in Eq. (1) for $CO_2$ was derived by comparing the priors generated with $\alpha = 1$ and $\beta = 0$ against profiles from the HIPPO (Wofsy, 2011) and ATom (Wofsy et al., 2018; Thompson et al., 2022) campaigns. We used $CO_2$ measurements from the Harvard quantum cascade laser spectrometer (QCLS) for HIPPO and $CO_2$ measurements from the NOAA Picarro for ATom. Only data from invididual vertical profiles (identified as data points where the `PFP/prof.no`

variable is $> 0$ in the merge files from https://doi.org/10.3334/ORNLDAAC/1581) with $\geq 10$ valid data points were used. The differences between the priors and observations below 800 hPa were averaged over $20°$ latitude bins and converted from units of ppm to multiples of the interannual $CO_2$ growth rate, derived from the MLO & SMO average deseasonalized trend. The output of the distance function $d$ (Eq. 2) was also averaged for all prior levels below 800 hPa and binned to $20°$ latitude bins.

The result is shown in Fig. 7a. The red line is a York fit (York et al., 2004) to the data using the inverse square of the standard

deviations of the prior-observation differences and distance function values in the latitude bins as the weights. This fit indicates setting $\alpha$ equal to -3.55 times the $CO_2$ interannual growth rate will give a latitudinal gradient that matches observations. Figure 7b shows the mean differences vs. latitude with $\alpha$ set to 1 (i.e. no adjustment) and with the best fit to the data. Using the $\alpha$ derived from Fig. 7a and $\beta = 0$, the priors show no latitudinal bias versus observations.

**Stratosphere:** $CO_2$ follows the algorithm laid out in Sect. 2.3. No additional modifications were required. For our purposes,

we assume that $CO_2$ DMFs are unaffected by stratospheric chemistry (e.g. $CH_4$ oxidation) and do not include a correction for chemistry in stratospheric $CO_2$.

## 3.3 $N_2O$

**Troposphere:** We set $\alpha$ in Eq. (1) to

$$\exp\left(\frac{-d}{\tau}\right) \tag{11}$$

where $d$ is the output of the distance function from Eq. (2) and $\tau = 121$ yr (the mean atmospheric lifetime of $N_2O$, following Myhre et al., 2013, Table 8.A.1). This imposes a slight additional north-south gradient to $N_2O$ in the troposphere.

**Stratosphere:** In the stratosphere, $N_2O$ is more complicated than $CO_2$ because it is removed, principally through photolysis forming nitrogen $N_2$ and an oxygen atom O, but also via a reaction with excited oxygen ($O(^1D)$) (Jacob, 1999). Andrews et al. (2001a) fit this loss of $N_2O$ in the lower stratosphere versus age of air with a third-order polynomial. We examined how this

polynomial compares to $N_2O$ data from the ACE-FTS instrument (Bernath et al., 2005) and found that the polynomial's skill in predicting the fraction of $N_2O$ remaining relative to the SBC ($F(N_2O)$) decreased above approximately 25 km altitude, with the polynomial overestimating the $N_2O$ mixing ratio by up to 150 ppb. We hypothesize this is due to different chemistry in the upper stratosphere compared to the lower stratosphere. As the original polynomial was based on lower stratospheric data,



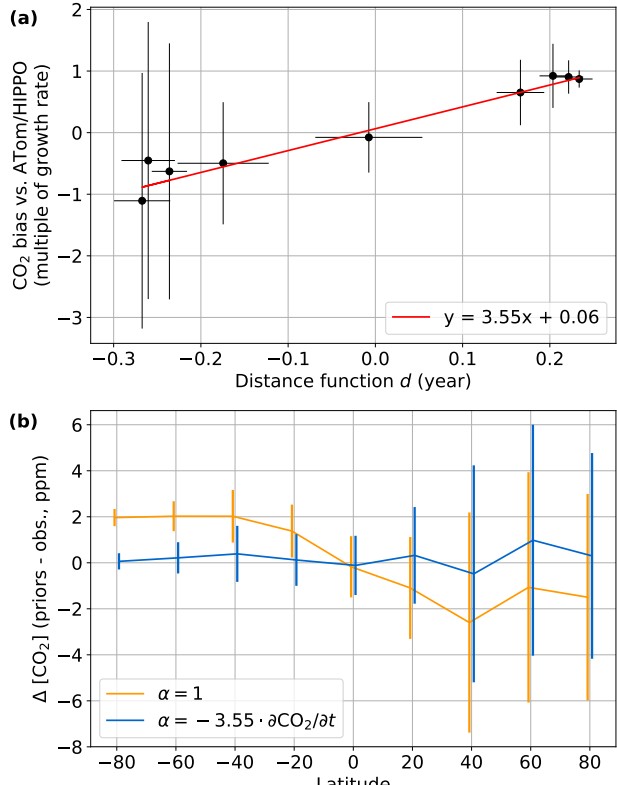

**Figure 7. (a)** Bias between the inital $CO_2$ DMFs and HIPPO/ATom profile vs. the distance function (Eq. 2) for profile levels below 800 hPa. Note that the $y$-axis is not in ppm, but in multiples of the interannual $CO_2$ growth rate. See text for details. **(b)** The mean difference between priors and observations in 20° latitude bins below 800 hPa, vs. latitude bin center. In both panels, error bars are $1\sigma$ standard deviations of the respective variable within the 20° latitude bins.

it did not capture this behavior. While the fraction of the $N_2O$ column in the upper stratosphere is small (a few percent above

20 km), our goal was to develop priors with reasonably accurate DMFs at all altitudes, not just where the bulk of the column mass is. Additionally, developing our own method to estimate $F(N_2O)$ allows us to be consistent when calculating the same quantity for $CH_4$ and HF.

We use $N_2O$ data from the ACE-FTS instrument to build a lookup table of the fraction of $N_2O$ remaining as a function of age of air and potential temperature. Strong et al. (2008) validated a previous version of the ACE-FTS $N_2O$ data and found

that mean differences between ACE-FTS and other stratospheric $N_2O$ measurements were $\pm10$ ppbv between 18 and 30 km, and mostly within $-2$ to $+1$ ppbv between 30 and 60 km. They note that these are large relative to the magnitude of $N_2O$ mole fractions at these altitudes; however, for our purposes, these are acceptable, given that we are averaging a large number of ACE-FTS profiles and need only a climatological relationship between fraction of $N_2O$ remaining, age of air, and potential temperature. Waymark et al. (2014) compared the version 3 ACE-FTS data (used in this work) to the version 2 evaluated by





Strong et al. (2008) and note that the main difference is a 10% reduction in $N_2O$ above 30 km. Thus the general results in Strong et al. (2008) should still hold. For ACE-FTS v3.5 data (one minor version earlier than that used in this work), Sheese et al. (2017) found biases between ACE-FTS and MIPAS (Michelson Interferometer for Passive Atmospheric Sounding) of between -9% to 5% and between ACE-FTS and MLS (Microwave Limb Sounder) of between -18% and 4% in the altitude range of 19 to 34 km.

To build the lookup table, age of air is computed as in Sect. 2.3; for each ACE profile, the stratospheric equivalent latitude is computed for the GEOS-5 FP-IT files that bound it in time, then it is interpolated to the latitude, longitude, and time of the profile. This equivalent latitude and the potential temperature calculated from ACE-FTS temperature and pressure is used as input to the CLaMS model from Sect. 2.3 to look up the age of air.

$F(N_2O)$ is defined relative to the stratospheric boundary condition in the ACE-FTS data, not the MLO & SMO record, to
ensure self-consistency and avoid introducing error from the bias between the ACE-FTS and MLO & SMO data (Fig. S9). The stratospheric boundary condition is computed from a quadratic fit in time of ACE-FTS $N_2O$ data in the tropics (latitude within $\pm 20°$) and with $360 \, \mathrm{K} < \theta < 390 \, \mathrm{K}$, excluding outliers (defined as values more than five times the median deviation from the median). This definition of the stratospheric boundary condition assumes that most of the air entering the stratosphere does so in the tropics and that the tropical tropopause is in that range of potential temperature values.

Finally, to compute the $F(N_2O)$ lookup table, the ACE-FTS data are binned by age of air (0.25 year increments) and potential temperature (variable increments; 50 K in the lower stratosphere to 200 K in the upper stratosphere). ACE-FTS data are excluded if:

   – $F(N_2O) < 0$,

   – altitude $\geq 70.0$ km (this is the top altitude in the TCCON priors),

– the profile is in the polar vortex.

Additionally, $F(N_2O)$ values $> 1$ are limited to 1. The resulting lookup table is shown in Fig. 8. As there are large gaps in age-$\theta$ space with no ACE-FTS data, we extrapolate to fill in these gaps. We use essentially a constant-value extrapolation along age, that is, if there is no value for a given age-$\theta$ bin, the nearest point at the same $\theta$ is used. Linear extrapolation along age is done second, using the nearest two points to determine the slope. In general, points in these extrapolated regions are expected
to be very infrequent, as the absence of ACE data suggests that those combinations of age and $\theta$ are rare in the atmosphere.

The need to capture how $F(N_2O)$ depends on both age and $\theta$ is apparent in Fig. 8. Consider the points in Fig. 8 at age = 5 years. Over the range of 1000 K, the $F(N_2O)$ decreases from $\sim 0.5$ to almost 0. This is likely because at greater $\theta$ (i.e. higher altitude) the $N_2O$ photolysis ($N_2O + h\nu \rightarrow N_2 + O$) pathway proceeds more rapidly than at lower altitudes. Age of air alone cannot capture this difference.





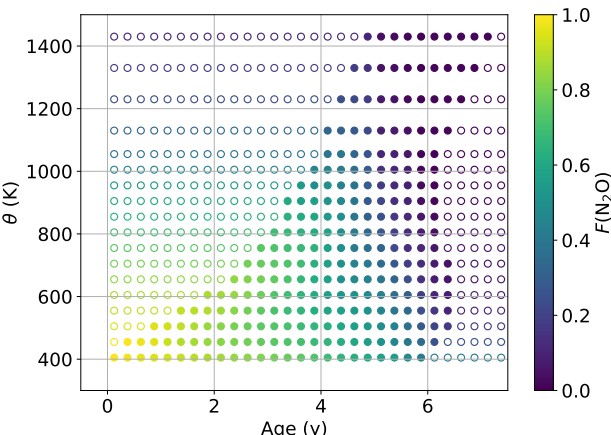

**Figure 8.** $F(N_2O)$ lookup table derived from ACE-FTS v3.6 data as a function of potential temperature and age of air. Unfilled circles are extrapolated points.

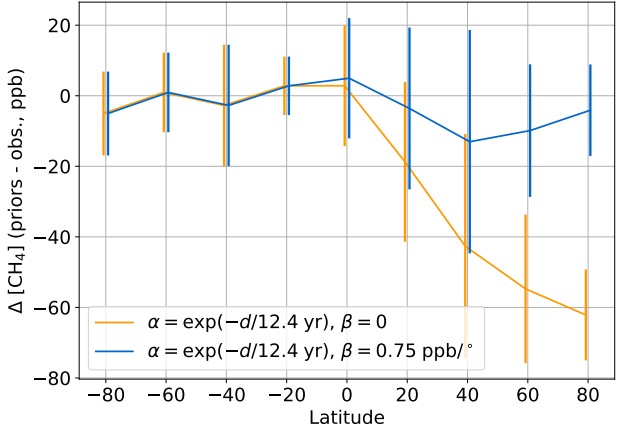

**Figure 9.** Differences in $CH_4$ between ATom and HIPPO observations and priors, binned as in Fig. 7b, with and without $\beta = 0.75 \, \text{ppb}/°$ correction in the northern hemisphere. Error bars are $1\sigma$ standard deviations in the $20°$ latitude bins.

### 3.4 CH$_4$

**Troposphere:** Similar to $N_2O$, the $CH_4$ priors use Eq. (11) as $\alpha$, with a lifetime of 12.4 yr (Myhre et al., 2013, Table 8.A.1). The orange line in Fig. 9 shows the mean prior vs. observation differences below 800 hPa in $20°$ latitude bins, as in Fig. 7b. A latitudinal bias in tropospheric methane mole fractions in the northern hemisphere remains. Therefore we set $\beta$ to $0.75 \, \text{ppb}/°$ in the northern hemisphere, which removes this bias (blue line, Fig. 9).





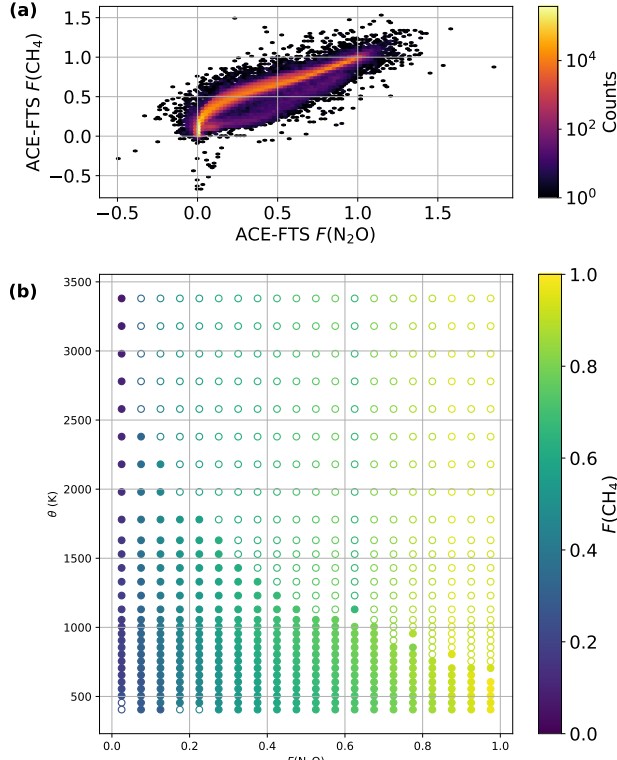

**Figure 10. (a)** 2D histogram of $F(N_2O)$ and $F(CH_4)$ from ACE-FTS. **(b)** The $F(CH_4)$ lookup table used in the GGG2020 algorithm. As in Fig. 8, filled circles are derived directly from data while the unfilled circles are extrapolated. $F$ values are computed as described in Sect. 3.3.

**Stratosphere:** $CH_4$ must also include a fraction remaining term, $F(CH_4)$, to account for stratospheric chemistry, similarly to $N_2O$. Figure 10a shows a tight correlation between ACE-FTS $N_2O$ and $CH_4$ in the stratosphere; therefore, we can use the relationship between $F(N_2O)$ and age derived in Sect. 3.3 as a basis for the $F(CH_4)$ lookup table.

  To compute the lookup table, we first limit the ACE-FTS data to points where $F(N_2O)$ and $F(CH_4)$ are positive, the $CH_4$ mole fraction is $< 2000$ ppb (points $\geq 2000$ ppb are almost certainly tropospheric), the profile is outside the polar vortex, and

the altitude is below 70 km. We bin the data by $F(N_2O)$ and $\theta$. Within each $F(N_2O)$ bin, outliers are rejected (distance $\geq 5\times$ median absolute deviation) and the mean $F(CH_4)$ value in each $F(N_2O)$ and $\theta$ bin pair is computed. As with $N_2O$, we use extrapolation to fill in parts of the lookup table not covered by ACE-FTS data. We use constant value extrapolation along the $\theta$ dimension first, then also along the $F(N_2O)$ dimension if necessary.

  To compute the stratospheric prior profiles, Eq. (8) is used with the $F(CH_4)$ value described above. To compute the $F(CH_4)$

value, the age and $\theta$ values are first used to compute the $F(N_2O)$ value as described in Sect. 3.3, and then the $F(CH_4)$ value is determined by linearly interpolating the lookup table in Fig. 10b to the required $F(N_2O)$ and $\theta$.





### 3.5 HF

Measurements of HF DMFs in the troposphere are very rare; the most recent direct measurement of gaseous fluoride that we found in the literature was Okita et al. (1974), which reported measurements around an aluminum refinery. Their measurements near but not downwind of the refinery reported fluoride concentrations of $< 1 \, \mu g \, m^{-3}$, or a DMF of order 10 to 100 parts per trillion (ppt). Spectroscopic measurements over Antarctica (Toon et al., 1989) and Switzerland (Zander et al., 1987) found upper tropospheric HF DMFs of 1 to 10 ppt were consistent with solar-viewing spectra.

For our purposes, we assume that the tropospheric DMF of HF is negligible compared to the stratospheric component, and so imposed a small but non-zero DMF of 0.1 ppt. This is less than the previous measurements (Okita et al., 1974; Zander et al., 1987; Toon et al., 1989), but the impact on HF retrievals should be small given that TCCON HF averaging kernels are usually $< 0.5$ below 200 hPa.

In the stratosphere, we once again make use of tracer-tracer relationships. HF is produced by reaction of fluorine atoms from photolysis of $COF_2$ and $COFCl$ (which are the products of destruction of CFC-11, CFC-12, and HFC-22) with $CH_4$, $H_2$, or $H_2O$ (Washenfelder et al., 2003). Thus, $CH_4$ and HF mole fractions are tightly anticorrelated in the stratosphere. Previous studies (e.g. Saad et al., 2014) have used this relationship to separate tropospheric and stratospheric $CH_4$ columns; here, we do the reverse, using $CH_4$ prior profiles to determine HF prior profiles.

We follow a similar approach to Saad et al. (2014); we determine the $CH_4$:HF slope ($m$) and directly compute the HF mole fraction from the $CH_4$ mole fraction as:

$$[HF] = \frac{[CH_4] - [CH_4]_{sbc}}{m} \tag{12}$$

where $[CH_4]_{sbc}$ is the $CH_4$ stratospheric boundary condition determined from the MLO & SMO record, as described in Sect. 2.3.2.

Because of the time dependence in the ratio of methane to the long-lived fluorine containing gases in the troposphere and because of the non-uniform ratio of the lifetime of $CH_4$ and the CFCs in the stratosphere, the slope $m$ depends on both time and latitude (Washenfelder et al., 2003; Saad et al., 2014). Before the beginning of the ACE-FTS data set in 2004, we use $CH_4$:HF slopes reported in Washenfelder et al. (2003). From 2004 on, we bin ACE-FTS $CH_4$ and HF data into the same three latitude bins (tropics, midlatitudes, and polar vortex) as for the age spectra (Sect. 2.3.2). We filter for $[CH_4] \leq 2000$ ppb and $[HF] \leq 10$ ppb and limit to altitudes $< 70$ km. The limit on $CH_4$ is imposed for the same reason as in Sect. 3.4; the limit on ACE-FTS HF is imposed due to erroneously large values of $\sim 200$ ppb found in rare cases (despite only using data with $CH_4$ and HF quality flags $\leq 1$). A 10 ppb upper limit was determined to only exclude these extraordinary values. The $CH_4$:HF slopes were fit as in Saad et al. (2014) using a robust fit with Tukey's biweighting function.

Finally we combine the ACE-FTS-derived slopes with those from Washenfelder et al. (2003) and fit the change over time with an exponential. This allows us to extrapolate forward or backward in time as needed. Each latitude bin has its own exponential fit that fits the bin-specific ACE-FTS slopes and the Washenfelder et al. (2003) slopes. (All bins used the same Washenfelder et al. (2003) data.) For consistency, we always take the slope from the exponential fit.



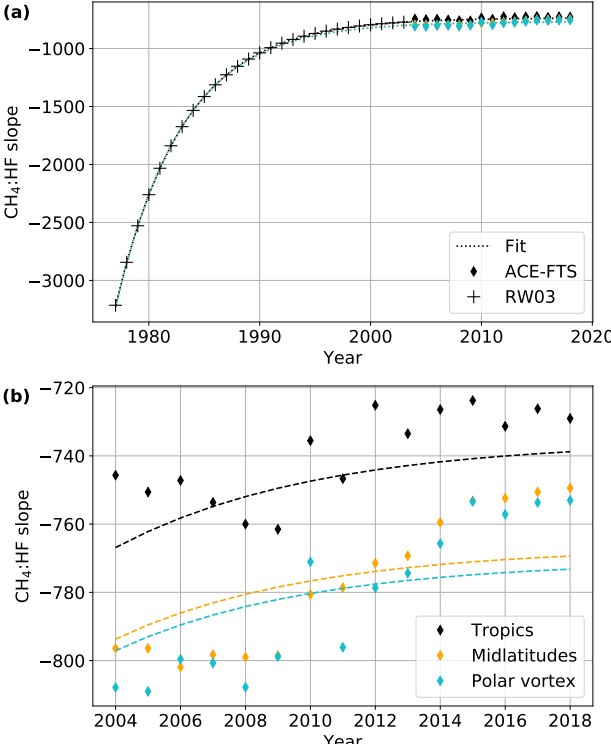

**Figure 11. (a)** $CH_4$:HF slopes and the exponential fits over the entire time period with data, be it from Washenfelder et al. (2003) (RW03) or ACE-FTS. **(b)** Similar to (a), but zoomed in on the ACE-FTS time period and colored by latitude bin.

Therefore, for each overworld level ($\theta \geq 380$ K), a $CH_4$ mole fraction is calculated (following Sect. 3.4) and the $CH_4$:HF slope for the year and latitude bin (based on equivalent latitude, Sect. 2.3) is used in Eq. (12) to compute the HF mole fraction. Note that we use the slope for the year of the observation, and not the year the air entered the stratosphere, because the slopes are based on observations for specific years.

### 3.6    CO

**Troposphere:** With a shorter tropospheric lifetime (of order months) than the above gases, CO requires a custom treatment in order to adequately account for its spatial variability. The GEOS-5 FP-IT product contains a CO forecast that shows reasonable skill in comparison to QCLS CO measurements taken during the ATom campaigns (Wofsy et al., 2018). We therefore adopt the GEOS-5 FP-IT CO product as the base profile for the CO priors with the following modifications.

First, our comparison against the first three ATom campaigns shows a low bias in the GEOS-5 FP-IT CO mole fractions,
as seen in Fig. 12a. While there is some variation with latitude, the pattern was not sufficiently clear to lend itself to a robust correction, therefore, we multiply the troposphere CO mole fractions by 1.23 ($= 1/0.81$) to bring them in line with ATom observations.





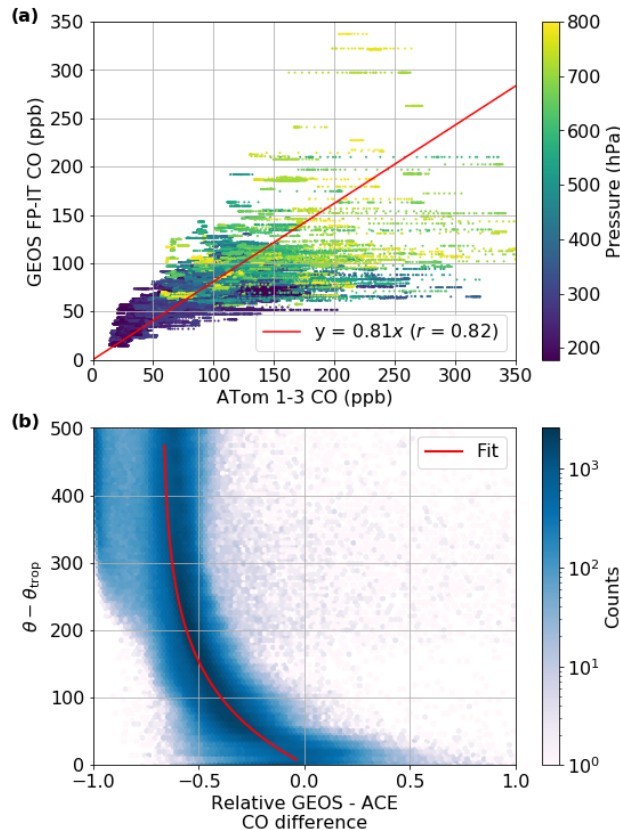

**Figure 12. (a)** Comparison of colocated ATom measured and GEOS-5 FP-IT forecasted CO mole fractions. GEOS-5 FP-IT CO matched to ATom observations using 4D nearest neighbor interpolation. The fit is a robust fit using a Tukey biweight function with no intecept. Only points with pressure < 800 hPa used. **(b)** Comparison of colocated ACE-FTS and GEOS-5 FP-IT CO data. The $y$-axis is potential temperature relative to the tropopause. The background shading is a 2D histogram of the relative bias between ACE-FTS and GEOS-5 FP-IT CO as a function of $\theta$; the red line is a fit through the mean bias.

**Stratosphere:** Comparison with ACE-FTS data in the lower stratosphere also demonstrates a low bias, which varies with altitude. However, the general structure is consistent as a function of potential temperature relative to the tropopause, as seen

in Fig. 12b. This can be represented by an exponential function.

Therefore, the overall CO correction has the form shown in Fig. 13. Below the tropopause, the 1.23 factor derived from ATom is used, while above 380 K (i.e. the stratospheric overworld) the exponential form derived from ACE-FTS is used. In the middleworld, we linearly blend between the two functions in order to provide a smooth transition.

The second correction required concerns the intrusion of mesospheric CO into the stratosphere. In the mesosphere, very

large mixing ratios of CO are produced through photolysis of $CO_2$. As this descends (especially in the polar vortex), it can lead to very large CO mole fractions at altitudes as low as 40 km. This process is not captured in the GEOS-5 FP-IT product,



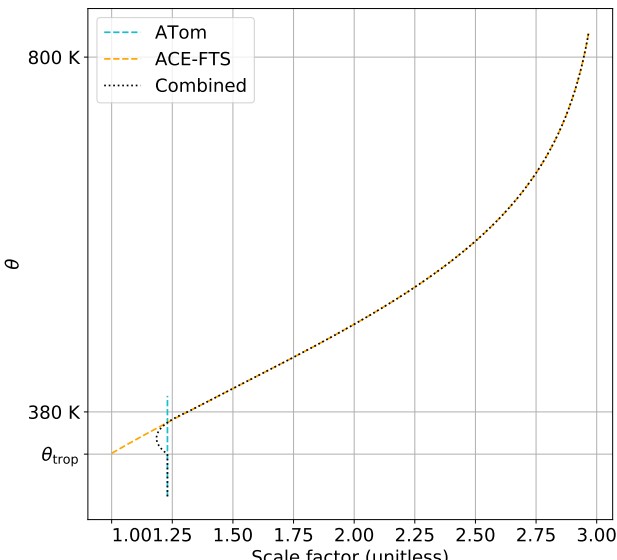

**Figure 13.** The form of the CO bias correction scaling factor. The blue and red lines show the form derived from ATom and ACE-FTS data, respectively while the black line shows the blending of these two corrections. Note that the ATom line is extended up to 380 K for reference, it does not imply that ATom collected data into the mid-stratosphere.

but is represented in the Canadian Middle Atmosphere Model (CMAM), which compares well with ACE-FTS and MLS data (Jin et al., 2009; Kolonjari et al., 2018). We use here output from a version of CMAM run with dynamics specified (see Sect. 2.2 of Kolonjari et al., 2018, and references therin).

Comparison of GEOS-5 FP-IT with ACE-FTS data shows the mesospheric CO impact beginning around 30 hPa and becoming dominant by 10 hPa. Therefore, we replace the GEOS-5 FP-IT CO with CMAM CO above 10 hPa (i.e. at pressure $< 10$ hPa) and linearly interpolate from GEOS-5 FP-IT to CMAM in pressure-log space between 30 and 10 hPa. The CMAM CO is drawn from a monthly climatology constructed from the monthly-averaged CO DMFs in the 30-year CMAM model run (available at http://climate-modelling.canada.ca/climatemodeldata/cmam/output/CMAM/CMAM30-SD/mon/atmosChem/

vmrco/index.shtml, last accessed 24 Jul 2019). CMAM model data before 2000 is not used in the climatology because there is not a trend present after 2000.

The third and final correction accounts for the mesospheric CO itself. While the priors used in TCCON retrievals have a 70 km ceiling, the CO above that altitude in the CMAM model can comprise up to $\sim 2.5\%$ of the total column, particularly in the polar regions. To account for this in the prior, we add an equivalent mass of CO to the top level of the priors. This is detailed in

Sect. S4 of the supplement.



## 3.7 H$_2$O and HDO

The H$_2$O profile is computed directly from the GEOS-5 FP-IT specific humidity. The HDO profile is directly computed from the H$_2$O profile as:

$$c_{\mathrm{HDO}} = c_{\mathrm{H_2O}} \cdot 0.14 \cdot [8 + \log_{10}(c_{\mathrm{H_2O}})] \tag{13}$$

where $c_{\mathrm{H_2O}}$ and $c_{\mathrm{HDO}}$ are the DMFs of H$_2$O and HDO, respectively. In the GGG retrieval, the line intensities of isotopologs are multiplied by the isotope abundance. This form therefore does not need to reproduce the abundance of HDO, but instead just the decrease of HDO relative to H$_2$O with altitude due to Rayleigh fractionation (Kuang et al., 2003). While reading the priors, GGG takes the absolute value of the HDO DMF to eliminate negative DMFs resulting from H$_2$O $< 10^{-8}$. In versions of ginput after 1.1.4, the absolute value of the HDO DMF is output.

## 460 4 Use as OCO-2/3 priors

The Orbiting Carbon Observatory 2 (OCO-2) and OCO-3 retrievals use these CO$_2$ priors starting in their respective version 10 products. The version 10 products use this algorithm exactly as described above except for one small change: in Eq. (1), $l$ is geographic, rather than effective, latitude. This difference ensures a smooth latitudinal variation in CO$_2$. Using effective latitude introduced discontinuities near the equator (Fig. S15a).

The specific structure of the discontinuities in Fig. S15a arise because version 10 of the OCO-2/3 algorithm uses an earlier version of the priors algorithm than GGG2020; in this earlier version, rather than transition between geographic latitude and effective latitude between 20° and 25°, effective latitude was used for profiles at all latitudes but disallowed from crossing the equator. (That is, a profile in the northern hemisphere could not have an effective latitude in the southern hemisphere and vice versa.)

Switching the version 10 priors to use geographic latitude for all soundings trades some ability to capture day-to-day variation in the troposphere for guaranteed spatially smooth priors (Fig. S15b), which is well worth it for nadir viewing instruments such as OCO-2 and OCO-3. In contrast, for discrete measurement sites such as TCCON, the ability to capture day-to-day variations is preferred.

The OCO-2/3 version 11 priors introduced an additional change to allow more frequent updating of the input in situ data. 475 GGG2020 and OCO-2/3 version 10 use a static file of MLO & SMO data as input that contains monthly averages of flask data prepared by NOAA (Dlugokencky et al., 2019) up through the end of 2018. These records are extended by extrapolation (see Sect. 2) as needed. This has the virtue of simplicity, but cannot capture anomalies in the trend of CO$_2$ such as those cause by El Niños.

The OCO-2/3 version 11 algorithm switched to using hourly in situ data from the continuous trace gas analyzers stationed 480 at MLO & SMO NOAA observatories (Thoning et al., 2021) that has undergone preliminary quality control, but not full background selection by NOAA personnel. These hourly in situ data are preprocessed by the priors code to produce monthly





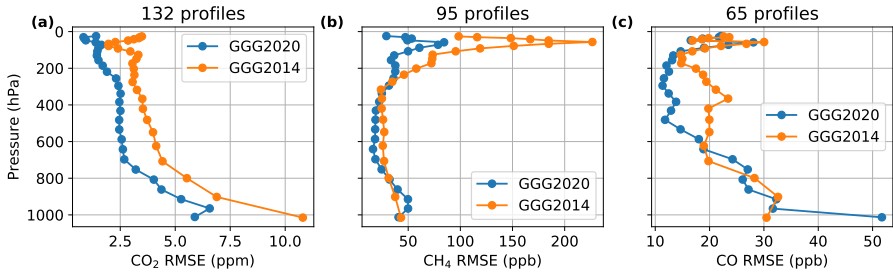

**Figure 14.** Root mean squared error (RMSE) of **(a)** $CO_2$, **(b)** $CH_4$, and **(c)** CO priors versus combined AirCore and aircraft observations. Data sources are listed in S1 and S2. In each panel, both the GGG2020 and GGG2014 priors' RMSE is shown. The number of profiles contributing to each panel is printed above the panel. FMI/RUG Sodankylä AirCore data above 20 km altitude are not included due to anomalously high mixing ratios in CO, $CO_2$ and $CH_4$ data above 20 km also excluded for consistency.

averages, allowing the main algorithm to use either monthly flask or hourly in situ data as needed. The preprocessing algorithm is described in Sect. S5 of the supplement.

# 5    Validation

## 5.1    Comparison with aircraft and AirCore observations

To directly validate the GGG2020 priors, we use aircraft data from the NOAA $CO_2$ GLOBALVIEWplus v5.0 Obspack (Co-operative Global Atmospheric Data Integration Project, 2019; Masarie et al., 2014), NOAA $CH_4$ GLOBALVIEWplus v2.0 ObsPack (Cooperative Global Atmospheric Data Integration Project, 2020; Masarie et al., 2014), and the Infrastructure for Measurement of the European Carbon Cycle (IMECC) campaign (Geibel et al., 2012), as well as AirCore (Tans, 2009; Karion et al., 2010) profiles from NOAA routine and campaign balloon flights (v20201223, Baier et al., 2021) and selected AirCore balloon flights from FMI/LSCE/RUG at the Sodankylä, Finland (Kivi and Heikkinen, 2016) and Nicosia, Cyprus (Messer-schmidt et al., 2012) TCCON sites. Data from tower measurements at Park Falls, WI, USA (Andrews et al., 2014; Desai et al., 2015), the Southern Great Plains Atmospheric Radiation Measurement facility near Lamont, OK, USA, and at the National Intitute of Water & Atmospheric Research Ltd. site in Lauder, New Zealand were used to extend airborne profiles in these locations to the surface as needed. The data used, and which gases are provided by each, are tabulated in Tables S1 and S2.

Figure 14 shows the root mean squared error (RMSE) for each vertical level of both the GGG2014 and GGG2020 priors. Mean and individual profile errors are given in Fig. S10. A breakdown of the number of profiles by gas and source is given in Table S4.

For $CO_2$, the RMSE is noticeably smaller at all altitudes for the GGG2020 priors compared to the GGG2014 priors (Fig. 14a). This results from removing a small but clear negative bias throughout the troposphere arising from an underestimate of the $CO_2$ secular growth rate in GGG2014. Using the MLO & SMO data eliminates that as a source of uncertainty for profiles





before 2019. (2019 is the first year that the MLO & SMO trend is extrapolated for GGG2020 as we chose to use a static file to avoid the complications of updating the input data in a reliable, reproducible manner, as discussed in Sect. 2.) In the stratosphere (above 200 hPa), the improved representation of stratosphere dynamics (Sect. 2.3) better captures the gradient of

$CO_2$ in the lower stratosphere, reducing the previous overestimate of lower stratospheric $CO_2$ in the GGG2014 priors.

The $CO_2$ RMSE for the GGG2020 priors is still greater near the surface than at higher altitudes. This may be due to the simplified seasonal cycle (Sect. 2.2). Comparing the priors to ATom and HIPPO observations in different seasons (Fig. S8) shows large differences near the northern hemisphere surface in spring and summer. As the seasonal cycle has latitudinal dependence, revising its parameterization will require adjustment to the distance function (Eq. 1) and the $\alpha$ and $\beta$ coefficients

(Table 3). This area will be revisited in a future version of the GGG priors.

$CH_4$ shows a small improvement in RMSE throughout most of the troposphere (Fig. 14b, 800 to 200 hPa). Above 200 hPa, the RMSE shows a greater improvement, again due to the improved representation of stratospheric dynamics. However, near the surface (below 800 hPa), the RMSE increases somewhat in the GGG2020 priors compared to the GGG2014 priors. This increase in RMSE is driven by near-surface $CH_4$ emissions not accounted for in the priors. Figure 15a shows differences

of the $CH_4$ priors vs. AirCore data (which has frequent sampling of areas with high emissions), colored by which TCCON site the prior represents. The bias in $CH_4$ below 800 hPa is clearly due to underestimated $CH_4$ in the Lamont, OK profiles. The Lamont TCCON site is situated near a region of significant oil and natural gas production (Karion et al., 2015), and thus experiences enhanced $CH_4$ mole fractions of 100 to 200 ppb near the surface (Fig. S13). Neither the GGG2014 nor GGG2020 priors attempt to account for local anthropogenic emissions. The increase in RMSE near the surface in the GGG2020 priors

is due to the removal of a compensating error in assumed vertical gradients—introducing the tropospheric effective latitude (Sect. 2.2.1) accounts for times when Lamont has a profile that varies less with altitude due to the influence of tropical air.

The GGG2020 CO priors' RMSE improves throughout the free troposphere (600 to 200 hPa). Unlike $CO_2$ and $CH_4$, RMSE is similar between GGG2014 and GGG2020 in the stratosphere (above 200 hPa). Near the surface, GGG2020 priors' RMSE is $\sim 20$ ppb greater than GGG2014. Figure 15b shows that this is driven by overestimated CO at the Armstrong Air Force Base

TCCON site and both over- and under- estimated CO at the Lamont TCCON site.

The cause of the over- and under- estimates in the Lamont profiles is not clear. The GGG2020 CO profiles are based on the CO field in the GEOS-5 FP-IT product (Sect. 3.6). The underestimated CO DMFs could be due to changes in energy economies in the region in recent years (Franklin et al., 2019; Willyard and Schade, 2019). GEOS-5 FP-IT uses 2008 anthropogenic CO emissions for all years after 2008 (Ott, private communication), so the CO priors would have no information on changes past

530    2008.

The overestimated CO at Armstrong is due to its proximity to Los Angeles. CO emissions in Los Angeles have been decreasing (Brioude et al., 2013), a trend not captured in GEOS-5 FP-IT as 2008 emissions are repeated for all years after 2008. Additionally, given that the GEOS-5 FP-IT model resolution is $0.67° \times 0.5°$ (longitude $\times$ latitude), the complex topography of the Los Angeles Basin, and that Armstrong is only $\sim 0.8°$ north of Los Angeles, the model is likely not able to capture the full

separation of Los Angeles and Armstrong profiles.



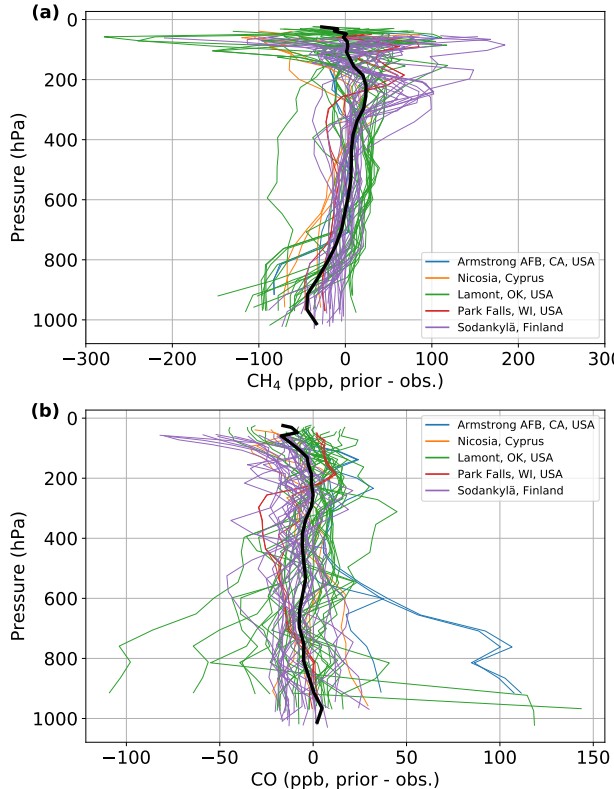

**Figure 15.** Difference plots for GGG2020 priors vs. **(a)** $CH_4$ and **(b)** CO AirCore data. The thinner, colored lines represent differences for individual profiles, the thick black line indicates the mean difference across all profiles shown. The individual differences are colored by their TCCON site.

Despite the increase in RMSE near the surface, overall the CO priors demonstrate important improvement. The scatter versus observations is noticeably reduced in the GGG2020 priors (Fig. S10) and the reduction in error in the mid-troposphere will be very beneficial to TCCON retrievals, as the CO averaging kernels increase with altitude up to the tropopause. Therefore, the retrievals are more sensitive to errors in the upper troposphere than the surface. We performed a sensitivity test where we
retrieved one year of XCO at Armstrong using two sets of priors. We found that the sensitivity of the retrieved XCO to the surface CO in the prior was small, only 0.024 ppb change XCO per 1 ppb change in surface prior CO (2.4%, Fig. S14c).

## 5.2   Indirect validation through retrievals

We can also evaluate the quality of the priors indirectly using the TCCON retrievals themselves. TCCON uses a scaling retrieval, in which the prior profiles are multiplied by scalar volume mixing ratio scale factors (VSFs) until the optimal match
between the forward spectroscopic model and measured spectrum is found. A VSF near 1 usually indicates that the prior profile represented the true atmospheric column abundance well (provided that the forward model spectroscopy is accurate), though it





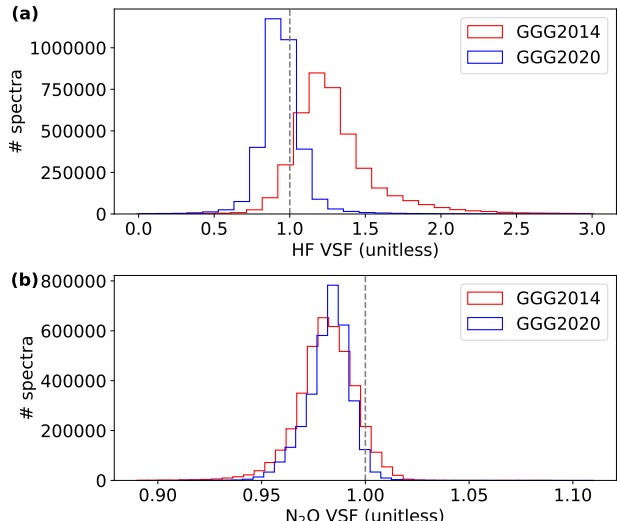

**Figure 16.** Volume mixing ratio (VMR) scale factors (VSFs) of **(a)** HF and **(b)**$N_2O$ retrieved using GGG2014 and a preliminary version of GGG2020. The vertical dashed gray line marks VSF = 1.

is also possible that compensating errors also yield a VSF near 1. However, given that the direct validation shown in Sect. 5.1 do not show compensating positive and negative biases on average, we expect such compensating errors are unlikely.

Figure 16 shows VSFs for HF and $N_2O$. Figure 16a shows that the median HF VSF decreased from $\sim 1.25$ in GGG2014 to $\sim 0.94$ in GGG2020, and the distribution is substantially tighter. HF is found only in the stratosphere (Washenfelder et al., 2003), therefore this result provides additional evidence that the stratosphere is well modeled by the GGG2020 priors.

Figure 16b shows that $N_2O$ VSFs moved slightly closer to 1 in GGG2020 with a tighter distribution. $N_2O$ is well mixed in the troposphere with an extremely uniform mixing ratio but varies substantially in the stratosphere due to loss via photolysis. Again, this implies improvement in the stratospheric priors and is a valuable check as we did not directly validate $N_2O$ against aircraft or AirCore observations due to sparse $N_2O$ profiles over TCCON stations.

Finally, we also consider the interhemispheric bias in $CH_4$ and $N_2O$ VSFs. For $CH_4$, Saad et al. (2014) found a $\sim 1\%$ bias between northern and southern hemisphere $CH_4$ VSFs using GGG2014 data, and Saad et al. (2016) determined that this was because the GGG2014 priors assumed a smooth DMF profile across the tropopause. In fact, the gradient in the lower stratosphere is driven by stratospheric circulation and $CH_4$ entering through the tropics (Sect. 2.3). As the priors now correctly account for this, the underlying error driving the interhemispheric bias in tropospheric $XCH_4$ in Saad et al. (2014) should now be eliminated and in fact the difference between median $CH_4$ VSFs between the northern and southern hemispheres has reduced by nearly 50% (Fig. S11).

For $N_2O$, the difference between median northern and southern hemisphere VSFs remains nearly the same magnitude ($\sim 0.4\%$, Fig. S12) but flips with the GGG2020 priors such that the median VSF is now greater in the southern hemisphere. Figure S12c compares the surface $N_2O$ DMFs from 6 NOAA stations against the surface DMFs in the priors for 5 TCCON





sites. While the priors' surface $N_2O$ in the southern hemisphere is approximately correct, there is a high bias in the northern hemisphere, possibly due to an incorrect assumed tropospheric lifetime (Sect. 3.3) or a need for an additional correction to our distance function (Sect. 2.2) that was not identified during development. This will be corrected in a future version of the TCCON priors.

## 6  Conclusions

GGG2020 introduces an improved algorithm to generate the prior profiles of $CO_2$, $N_2O$, $CH_4$, HF, CO and other gases needed for TCCON retrievals. The versions 10 and 11 OCO-2 and OCO-3 retrievals also use these $CO_2$ profiles. This approach is specifically designed to account for variations in vertical profiles due to synoptic-scale latitudinal motion of airmasses. Direct validation against aircraft and AirCore observations shows consistent reduction of error in the free troposphere and lower stratosphere, and indirect validation by examining the magnitude of retrieved TCCON VSFs gives further evidence that the accuracy of the priors in the stratosphere has improved.

An important guiding principle for the GGG2020 priors algorithm was to limit, as much as possible, dependence on ongoing measurements or models. Doing so means that retrievals using these priors produce data that can be treated as statistically independent with most existing and future measurements and models. Only $CO_2$, $CH_4$, and $N_2O$ measurements from the Mauna Loa and American Samoa observatories and CO from the GEOS FP-IT model system are directly ingested, so direct comparisons of TCCON GGG2020 or OCO-2/3 data with these data sources would not be not fully independent. As latitudinal gradients from the HIPPO and ATom campaigns and correlations of $N_2O$, $CH_4$, and HF from the ACE-FTS instrument are used as well, comparisons between TCCON or OCO-2/3 and HIPPO, ATom, or ACE-FTS data should note that correlations of these specific characteristics (i.e. latitudinal gradients, $N_2O$/$CH_4$/HF correlations) are correlated by design.

There remain areas for improvement. The age of air parameterization used in the troposphere is known to underestimate the age of air compared to $SF_6$ measurements and anthropogenic emissions are not accounted for except in the CO priors. Addressing these issues is planned for a future version of GGG; at that time, we will evaluate whether incorporating additional data from measurements or models produces worthwhile improvements in the priors' accuracy. Nevertheless, this represents a significant improvement for the GGG2020 TCCON retrieval.

*Code and data availability.* The code to generate GGG2020 prior profiles is the "ginput" package, available from GitHub (Laughner, 2022). GGG2020 TCCON data uses ginput version 1.0.6, which is scientifically identical to the publicly archived 1.0.7 version (Laughner et al., 2021). HIPPO data was obtained from https://data.eol.ucar.edu/cgi-bin/codiac/fgr_form/id=112.123/agree. ATom data was obtained from https://doi.org/10.3334/ORNLDAAC/1581. Obspack aircraft data ($CO_2$ GLOBALVIEWplus v5 and $CH_4$ GLOBALVIEWplus v2 were obtained from https://www.esrl.noaa.gov/gmd/ccgg/obspack/. NOAA AirCore data (v20201223) was provided by Bianca Baier and Colm Sweeney. Sodankylä AirCore data was provided by Huilin Chen and Rigel Kivi. Nicosia AirCore data was provided by Pierre-Yves Quehe. The CMAM model data use in the CO priors was downloaded from https://climate-modelling.canada.ca/climatemodeldata/cmam/



output/CMAM/CMAM30-SD/mon/atmosChem/vmrco/index.shtml (last access 24 Jul 2019). ACE-FTS v3.6 data are available fromhttps: //databace.scisat.ca/level2/; access to these products requires registration.

*Author contributions.* JLL created the priors code, carried out the validation, and led the writing of the manuscript. SR developed the code to read GEOS-FPIT meteorology and interpolate to TCCON locations. MK assisted with development of the $CO_2$ priors. GCT developed the original GGG priors, of which the climatological profiles used for the secondary gases (Sect. 2.4), seasonal cycle parameterization, and tropospheric distance function are retained in this work. POW guided the overall project. Other authors contributed data for validation of the priors. All authors reviewed the manuscript.

*Competing interests.* One coauthor is a member of the AMT editorial board.

*Acknowledgements.* The authors are deeply grateful to Arlyn Andrews from NOAA for providing the approach to generate the stratospheric $CO_2$ which was adapted and expanded upon for the other stratospheric priors, as well as the tables of stratospheric age-of-air used throughout this work. JLL and POW acknowledge funding from NASA grant NNX17AE15G. We thank all TCCON partners for carrying out GGG2014 and GGG2020 retrievals that provided the VSFs for Sect. 5.2. The AirCore campaign in Cyprus received support by the European Union's Horizon 2020 research and innovation programme under grant agreement No. 856612 and the Cyprus Government and additional support from the Service National d'Observation ICOS-France-Atmosphere coordinated by LSCE. The TCCON Nicosia site has received additional support by the European Union's Horizon 2020 research and innovation programme under grant agreement No. 856612 and the Cyprus Government, and the University of Bremen. The $CH_4$ tower measurements at Park Falls were supported by the DOE Ameriflux Network Management project award to the ChEAS core site cluster, NSF #0845166 and NSF #1822420. NOAA/GML AirCore profiles were supported by NASA grant 80NSSC18K0898. The authors are grateful to Peter Bernath for his leadership of the ACE-FTS project, which was invaluable to this work. The Atmospheric Chemistry Experiment (ACE), also known as SCISAT, is a Canadian-led mission mainly supported by the Canadian Space Agency. A portion of this research was carried out at the Jet Propulsion Laboratory (JPL), California Institute of Technology, under a contract with NASA (80NM0018D0004). Government sponsorship acknowledged.



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
