# Peer review of "A new algorithm to generate a priori trace gas profiles for the GGG2020 retrieval algorithm"

_Atmospheric Measurement Techniques, 2022_

## Author Comment (AC1)

**Paper title**

**Response to Anonymous Referee #2**

Joshua L. Laughner on behalf of all coauthors

January 10, 2023

We thank the reviewer for their time in evaluating our manuscript, especially given its length, and are excited by the positive responses that the paper received. Below we respond to the individual comments. The reviewer's comments will be shown in red, our response in blue, and changes made to the paper are shown in black block quotes. Unless otherwise indicated, page and line numbers correspond to the original paper. Figures, tables, or equations referenced as "R$n$" are numbered within this response; if these are used in the changes to the paper, they will be replaced with the proper number in the final paper. Figures, tables, and equations numbered normally refer to the numbers in the original discussion paper.

The paper is well written. It will become more comprehensive if authors add a brief description of "how much does the new algorithm improve the TCCON retrieval quantitatively" in the abstract and/or conclusion even though required accuracy is described in the introduction and the supplement.

We have added a new figure (Fig. 17 in the revised paper) that shows histograms of how the bias in the TCCON retrieved quantities versus AirCore profiles changes with the new priors. The implications of this figure are discussed in a new subsection (§5.3). We have also added some text to the conclusion that refers back to this subsection:

> "The column-average mole fractions retrieved by TCCON shift relative to in situ column averages by up to 0.2 ppm for $CO_2$, 13 ppb for $CH_4$, and 1 ppb for CO. For the standard TCCON $CO_2$, $CH_4$, and experimental $lCO_2$ ($CO_2$ with stronger sensitivity to the surface) products the new priors produce an overall improvement relative to in situ column averages. The CO and experimental $wCO_2$ (stronger sensitivity to the upper atmosphere) products compare slightly worse overall to in situ data using the new priors. For CO, this is likely due to overestimated anthropogenic CO emissions in the source model. Finding a way to correct this, either by using a different model run or applying a geographically-varying correction, will be a high priority for the next version of the TCCON priors. The reason for the slight worsening of the $wCO_2$ retrievals is not yet clear."

Line 83, "Negatively impact", Definition of "negative" or more detailed explanation will improve reader's understanding.

We have expanded this paragraph into two to better separate the (enlarged) explanation of when extrapolation errors impact the retrieval from our decisions whether to accept or address said error:

"Errors in extrapolating the MLO & SMO DMFs will negatively impact the TCCON retrievals if the error in extrapolation introduces an error in the profile shape, due to an El Niño year, for example. **In a scaling retrieval, such as the GGG algorithm used by TCCON, the posterior optimal profile is the prior profile multiplied by a scale factor, with the same scale factor applied to all levels. At its core, the algorithm we are describing here builds the priors by calculating what date to pull the MLO & SMO DMFs from for each level in the prior. If the extrapolation error caused all the MLO & SMO DMFs to be incorrect by the same percentage, this would manifest as the prior profile being incorrect by that percentage, for which a scaling retrieval can theoretically perfectly account. However, if the error in MLO & SMO DMFs is not the same for each level in the prior, that means the error in the prior cannot be represented by the same scalar multiplier for every level, and so a scaling retrieval could never completely eliminate the error in the posterior profile.**"

"Currently, we estimate the error in the MLO & SMO DMFs due to extrapolation to be about 0.25% for $CO_2$, 0.15% for $N_2O$, and 0.6% for $CH_4$ over a five-year extrapolation (see S2 in the supplement for details). We deem this level of uncertainty acceptable for TCCON priors. **How errors in the priors alias into the posterior state in a profile retrieval, such as that used by OCO-2 and -3, is more complex. However, the OCO-2/3 retrieval uses a relatively tight covariance matrix for levels in the stratosphere (see Fig 3-15 of Crisp et al., 2021), making it important that the priors not exhibit any long-term drift in these levels.** Therefore, when these priors are used for the version 11 OCO-2/3 retrievals, more recent NOAA data is ingested (see Sect. 4)."

Section3, ACE-FTS data. Even though number of data is limited and measurement of the lower troposphere is difficult, vertical profile from solar occultation data are accurate. What is the lower -altitude limit for the application of the paper?

We use ACE-FTS data where the potential temperature is $\geq 380$ K (as calculated from the ACE-FTS pressure and temperature profiles). This is not due to concerns over the ACE data, but rather because our algorithm only uses ACE-derived quantities for levels in the stratospheric overworld, defined by $\theta \geq 380$ K, thus we elect to be consistent. This has been added to the bullet point list of reasons why ACE data is filtered out in Sect. 3.3.

Section 5.1 Validation with Air CORE. Description of ideal numbers of AirCore measurement at each site and discussion on ideal global distributions of AirCore sites will be helpful for readers.

While I see how this could certainly help inform future plans to expand the AirCore program, it's not clear to me how this would add to the readers' understanding of the validation methodology or results. Further, to answer this rigorously would probably need some sort of OSSE-like study, with a given truth and uncertainty. Given that, I would prefer to keep this section focused on validating the new priors with what data is available, especially given the already substantial length of the manuscript.

Lines 53 and 202 "Potential Vorticity". Potential Vorticity (PV) appears twice.
Second instance changed to just "PV".

Line 524 the Armstrong Air Force Base TCCON site, Lines 531, 535 Armstrong, Figure 15 Armstrong AFB, Line 524 "the Armstrong Air Force Base (AFB) TCCON site"? Lines 531, 534 Armstrong AFB.
Changed as suggested.

**References**

Crisp, D., O'Dell, C., Eldering, A., Fisher, B., Oyafuso, F., Payne, V., Drouin, B., Toon, G., Laughner, J., Somkuti, P., McGarragh, G., Merrelli, A., Nelson, R., Gunson, M., Frankenberg, C., Osterman, G., Boesch, H., Brown, L., Castano, R., Christi, M., Connor, B., McDuffie, J., Miller, C., Natraj, V., O'Brien, D., Polonski, I., Smyth, M., Thompson, D., and Granat, R.: Orbiting Carbon Observatory (OCO-2) Level 2 Full Physics Algorithm Theoretical Basis Document, Version 3.0 - Rev 1, `https://docserver.gesdisc.eosdis.nasa.gov/public/project/OCO/OCO_L2_ATBD.pdf`, 2021.

---

## Author Comment (AC2)

**Paper title**

**Response to Anonymous Referee #1**

Joshua L. Laughner on behalf of all coauthors

January 10, 2023

We thank the reviewers for their time in evaluating our manuscript, especially given its length, and are excited by the positive responses that the paper received. Below we respond to the individual comments. The reviewer's comments will be shown in red, our response in blue, and changes made to the paper are shown in black block quotes. Unless otherwise indicated, page and line numbers correspond to the original paper. Figures, tables, or equations referenced as "R$n$" are numbered within this response; if these are used in the changes to the paper, they will be replaced with the proper number in the final paper. Figures, tables, and equations numbered normally refer to the numbers in the original discussion paper.

The comparison result from CO in Section 5.1 is a little bit disappointing, but not surprising given that CO is very variable.... I would like to see two extra comparisons that can be done quickly. One is the improvements of GGG2020 compared to GGG2014 after removing Armstrong and Lamont sites; The other is the comparison between GEOS-5 CO with other observation-assimilated CO simulations (for example, the CAMS global atmospheric composition forecast model, link below) over these two sites to see if the CO overestimate/underestimate is heavily model-dependent.

We have added these as Figs. S15 and S16 in the revised supplement. These comparisons show that the CO profiles are reasonable outside of urban areas or other locations with substantial fossil fuel combustion. These implications are discussed in two paragraphs added to Sect. 5.1:

> "Outside of urban or energy-intensive locations, the agreement between the new GGG2020 priors and co-located in situ profiles is much improved. Figure S15 compares RMSEs and mean prior vs. in situ differences for CO when Armstrong AFB, Lamont, and Orléans (another near-urban location) are excluded from the comparison. In that case, the RMSE reduces by about a factor of two or better at all levels except the surface in the new GGG2020 priors compared to the GGG2014 priors."

> "We compared CO profiles from the GEOS FP-IT product to the Copernicus Atmospheric Monitoring Service (CAMS) model to see if this issue of overestimated CO is common among models. The results for 2018 through 2022 are shown in Fig. S16. In general, GEOS FP-IT CO is dramatically greater than CAMS CO

in Los Angeles (at the Pasadena TCCON site). This is also true at Armstrong AFB, but to a lesser extent. In Paris, both models exhibit very high surface CO on some of the sampled days, though this was more frequent in the GEOS FP-IT CO profiles. At Lamont and East Trout Lake, both models had CO DMFs of similar magnitude (even with our factor of 1.23 scaling applied to the GEOS FP-IT data), with the main difference in vertical distribution. While the factor of 1.23 applied to bring the GEOS FP-IT CO in line with ATom observations (Fig. 12) definitely aggravates the GEOS FP-IT overestimate in urban areas, it improves the mean CO in more remote areas. In the future, drawing CO profiles from a model that better represents urban-rural CO gradients would improve the CO priors, but requires an existing model run that also covers the full range of times needed by TCCON (from 2004 on)."

Line 44: The description of the "1%" error in the shape of CO2 is a little bit ambiguous to me, and can be more specific, although you already have supplementary materials to explain that. One or two more sentences to explain that in the main text may help. Also, a change in the lower troposphere by 4ppm is not the same as changing the XCO2 by 1%. Similarly, the different scenarios in Figure S1 may represent different changes in XCO2, which may be confusing when compared to the retrieval error of ¡0.025% which is for XCO2.

We have edited this paragraph to more explicitly define what a 1% error in shape means in this context by specifying that it means the error in the prior relative to some truth changes from one level to the next:

"The relationship between the shape error in the prior and the error in the retrieved column amount depends on the averaging kernels. For TCCON $CO_2$ retrievals, testing with synthetic spectra shows that a **4 ppm error in the profile shape (defined as the error in the prior compared to the true profile changing by $\pm 4$ ppm between the top and bottom levels)** leads to an error of $\leq 0.025\%$ in $XCO_2$ at solar zenith angles (SZAs) $\lesssim 60°$, and $\leq 0.125\%$ up to SZA $\approx 75°$. (Details of how this was quantified are given in Sect. S1.) This means that for typical SZAs observed by TCCON, an error of **about 4 to 8 ppm** in the $CO_2$ prior results in a retrieval error well below the 0.25% ceiling required for TCCON data."

As for the different scenarios in Fig. S1 representing different $XCO_2$ values, that was intentional. The results shown in Fig. S1 are intended to test the retrieval sensitivity to errors in both shape and total $CO_2$ column. We have made this explicit in the supplement:

"Figure S1 shows the different prior profiles (panel a) and the resulting change in retrieved $XCO_2$ compared to the true profile (panel b). We defined two types of shape error: a "jump" where the $CO_2$ DMF increases or decreases suddenly at a specific altitude, and a "linear" error where the $CO_2$ DMF varies linearly with respect to pressure. For all shape errors, we defined a 1% error to mean that the DMF changes by 1% (4 ppm) between the top and bottom of the profile. Both the "jump" and "linear" cases each have three subcases that vary whether

the troposphere, stratosphere, or both have the error. **These various profiles represent different errors in both the shape *and* prior $XCO_2$ values. This was deliberate to test how the retrieval is sensitive to not only the error in shape but the total amount of prior $CO_2$.**"

Line 6: "improving the description of CO2, CH4, N2O, HF, and CO in the stratosphere", please rephrase.
Changed to:

"A particular focus of this work is improving the **accuracy of $CO_2$, $CH_4$, $N_2O$, HF, and CO across the tropopause and into the lower stratosphere.**"

Figure 12: Please add a unit (or is it just fraction/ratio?) for the x-axis of the lower panel.
This is the unitless relative difference between GEOS and ACE CO, we have clarified this in the caption:

"Comparison of colocated ACE-FTS and GEOS-5 FP-IT CO data. **The $x$-axis is the unitless relative difference, (GEOS - ACE)/ACE.** The $y$-axis..."

Figure 12: In the caption: The fit is a robust fit using a Tukey biweight function with no intercept. Please explain "Tukey biweight function". Also "intecept" is a typo.
Clarified as:

"The fit is a robust fit using a Tukey biweight function with no intercept, **i.e. using the `RLM` linear model with `M = TukeyBiweight()` from the Python `statsmodels` package (Seabold and Perktold, 2010).**"

**References**

Seabold, S. and Perktold, J.: statsmodels: Econometric and statistical modeling with python, in: 9th Python in Science Conference, 2010.